# Ex vivo expansion potential of murine hematopoietic stem cells is a rare property only partially predicted by phenotype

Qinyu Zhang*, Rasmus Olofzon, Anna Konturek-Ciesla, Ouyang Yuan, David Bryder*

Division of Molecular Hematology, Department of Laboratory Medicine, Lund Stem Cell Center, Faculty of Medical, Lund University, Lund, Sweden

*For correspondence:
qinyu.zhang@med.lu.se (QZ);
david.bryder@med.lu.se (DB)

**Competing interest:** The authors declare that no competing interests exist.

**Abstract** The scarcity of hematopoietic stem cells (HSCs) restricts their use in both clinical settings and experimental research. Here, we examined a recently developed method for expanding rigorously purified murine HSCs ex vivo. After 3 weeks of culture, only 0.1% of cells exhibited the input HSC phenotype, but these accounted for almost all functional long-term HSC activity. Input HSCs displayed varying potential for ex vivo self-renewal, with alternative outcomes revealed by single-cell multimodal RNA and ATAC sequencing profiling. While most HSC progeny offered only transient in vivo reconstitution, these cells efficiently rescued mice from lethal myeloablation. The amplification of functional HSC activity allowed for long-term multilineage engraftment in unconditioned hosts that associated with a return of HSCs to quiescence. Thereby, our findings identify several key considerations for ex vivo HSC expansion, with major implications also for assessment of normal HSC activity.

## eLife assessment

This study presents a **valuable** dissection on how functional HSCs are expanded in PVA cultures. The functional and multi-omic analyses provided are **convincing**, although the additional data and their analysis provided during revision could have been included in the test to assist readers and to strengthen the published manuscript. Nevertheless, the present work will be of value for stem cell biologists interested in HSC regulation.

## Introduction

Hematopoietic stem cells (HSCs) are the functional units in bone marrow transplantation (BMT) and gene therapy approaches via their ability for multilineage differentiation and long-term self-renewal (*Konturek-Ciesla and Bryder, 2022*). However, restrictions in HSC numbers, the requirement of harsh host conditioning, and challenges with how to genetically manipulate HSCs with retained function are barriers for BMT/gene therapy (*Walasek et al., 2012*). Due to the lack of reliable models for accurate assessment of human HSCs, mice continue to be the preferred experimental model for studying HSC biology. However, studying murine HSC biology also poses challenges due to the scarcity of HSCs.

Recently, culture conditions supporting efficient maintenance and expansion of murine HSCs were reported (*Wilkinson et al., 2019*). However, the expansion of functional in vivo HSC activity over a 28-day period (estimated to 236- to 899-fold) was not on par with the total cell expansion (>$10^4$-fold), demonstrating that large amounts of differentiated progeny were generated along with self-renewing

HSC divisions (*Wilkinson et al., 2019*). Moreover, candidate HSCs (cHSCs) by phenotype represented only a minor portion of the cells at the culture endpoint (*Wilkinson et al., 2019*), although assessment of HSC activity based on this might be obscured by previous observations that HSCs change some phenotypes in culture (*Zhang and Lodish, 2005*). While in vitro differentiation complicates assessments of HSC expansion, it might also be beneficial. For example, while HSCs can support life-long hematopoiesis after BMT, they have limited capacity to rescue the host from lethal conditioning (*Nakauchi et al., 1999*). To address this limitation, there is a requirement for progenitor cells that can efficiently supply the host with mature blood cells, such as erythrocytes and platelets (*Yang et al., 2005*; *Na Nakorn et al., 2002*).

When transplanted, in vivo HSC activity is typically determined using both quantitative and qualitative parameters (*Purton and Scadden, 2007*). Most commonly, this is achieved by comparing to a known amount of competitor cells (*Harrison et al., 1993*). However, although the competitive repopulation assay (CRA) represents a mainstay in experimental HSC biology, it associates with shortcomings often neglected. First, while clinical BMT aims at transplanting large numbers of HSCs (*Barnett et al., 1999*; *Trébéden-Negre et al., 2010*), experimental murine BMT typically use limited HSC grafts, which can strongly influence on HSC function (*Säwén et al., 2016*). Second, the CRA is from a quantitative perspective rather blunt, with sigmoidal rather than linear characteristics (*Harrison et al., 1993*). This limits the boundaries of the effects that can be measured. Third, the CRA assesses HSC activity based on overall hematopoietic chimerism, which might reflect poorly on the actual HSC activity given the vastly different turnover rates of different blood cells lineages (*Bryder et al., 2004*). At the same time, alternative assays such as limit dilution assays or single-cell transplantation assays are cumbersome, expensive, and associated with ethical concerns. Newer developments in DNA barcoding technology have overcome many of these issues (*Lu et al., 2011*).

Another concern with BMT is the use of aggressive host conditioning to permit engraftment. While eradicating malignant cells is an important part of treatment regimens for leukemic patients, the fates of HSCs in conditioned hosts versus native hematopoietic contexts are dramatically different (*Busch and Rodewald, 2016*). But transplantation into unconditioned recipients is inefficient and requires large amounts of transplanted HSCs to saturate available niches (*Brecher et al., 1982*; *Bhattacharya et al., 2009*). To achieve even limited HSC chimerism in unconditioned hosts can have strong long-term therapeutic effects (*Bhattacharya et al., 2006*) and many experimental approaches might benefit from avoiding host conditioning.

Here, we provide molecular and functional details on HSC differentiation and self-renewal following culture and define a set of parameters critical for in vivo assessment of HSC activity. Importantly, we find that HSC heterogeneity was a key influencing factor for HSC expansion even among rigorously purified cHSCs. This determinism has broad implications for experiments involving in vitro manipulation of HSCs, but also when assessing normal HSC function by BMT.

## Results

### In vivo HSC activity is restricted to cells expressing high levels of EPCR

We first assessed the in vivo reconstituting activity of cHSCs with different phenotypes. The endothelial protein C receptor (EPCR or PROCR/CD201) is highly expressed on HSCs (*Balazs et al., 2006*), while CD41 (ITGA2B) was suggested to be functionally relevant for HSC maintenance and homeostasis (*Gekas and Graf, 2013*). Based on expression of EPCR and CD41, we isolated four subfractions within BM SLAM LSK cells (Lin⁻Sca1⁺cKit⁺CD150⁺CD48⁻/low) (*Figure 1A*) and competitively transplanted them at equal numbers into lethally irradiated recipients (*Figure 1B*). The reconstitution from the three EPCR^high fractions was restricted to the peripheral blood (PB) myeloid lineages 4 weeks post-transplantation (*Figure 1C*), and most of these recipients displayed robust long-term (beyond 16 weeks) multilineage reconstitution with equivalent cHSC chimerism in the BM (*Figure 1C–D*). EPCR^high CD41⁻ cells produced slightly more lymphoid cells than the EPCR^high CD41⁺ cells at the endpoint (*Figure 1—figure supplement 1A*), but the cHSC chimerism in the BM was very similar to other tested EPCR^high fractions (*Figure 1D*). In contrast to *Gekas and Graf, 2013*, we failed to observe that staining with the CD41 antibody (MWreg30 clone) compromises HSC engraftment. Transplantation of EPCR negative cells failed to associate with either long-term PB or BM-cHSC chimerism, demonstrating lack of HSC activity within this fraction (*Figure 1C–D*).

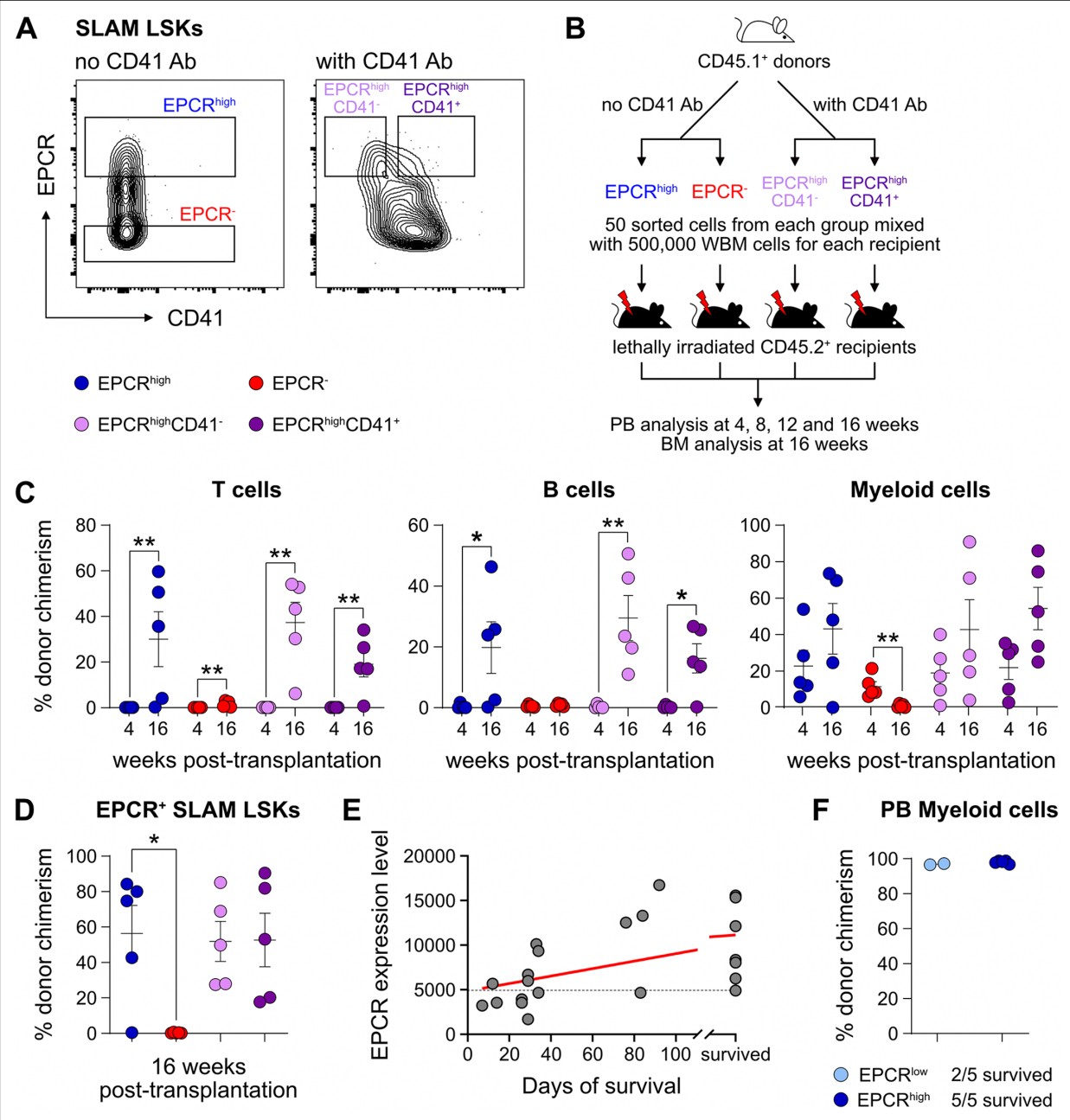

**Figure 1.** Endothelial protein C receptor (EPCR) expression within the bone marrow (BM) SLAM LSK fraction is a high-confidence predictor of transplantation-associated hematopoietic stem cell (HSC) activity. (**A**) Expression patterns of EPCR and CD41 within the BM LSK SLAM fraction. Gates depict the assessed cell fractions. (**B**) Strategy used to assess the correlation of EPCR and CD41 expression to the in vivo HSC activity. (**C**) Test cell-derived chimerism in peripheral blood (PB) 4 and 16 weeks post-transplantation. n=5 per group. (**D**) Test cell-derived chimerism in BM EPCR$^+$ SLAM LSKs 16 weeks post-transplantation. n=5 per group. (**E**) Correlation between duration of radioprotection and EPCR expression levels. BM LSK SLAM cells were co-stained with EPCR and index-sorted at one cell per well, cultured for 21 days, and the content of each well transplanted to individual recipients (n=22). Correlation to mortality of individual mice was made by assessing which well was transplanted into which mouse and coupling this to the index-sorting information. The gray dash line indicated the separation of EPCR higher (>4900) or lower expression (<4900) cHSCs. (**F**) Donor contribution in PB myeloid cells. Mice were transplanted with ex vivo expanded cells from either 50 SLAM LSK EPCR$^{low}$ (n=5) or 50 SLAM LSK EPCR$^{high}$ cells (n=5) and assessments made 16 weeks after transplantation. All data points depict values in individual recipients. Error bars denote SEM. The asterisks indicate significant differences. *, p<0.05; **, p<0.01. In (**E**) a regression line was generated based on an endpoint survival of 150 days (the time at which the experiment was terminated).

The online version of this article includes the following source data and figure supplement(s) for figure 1:

**Source data 1.** Raw data for *Figure 1C*: Donor chimerism in peripheral blood (PB) 4 and 16 weeks post-transplantation.

*Figure 1 continued on next page*

*Figure 1 continued*

**Source data 2.** Raw data for *Figure 1D*: Donor chimerism in bone marrow (BM) EPCR$^+$ SLAM LSKs 16 weeks post-transplantation.

**Source data 3.** Raw data for *Figure 1E*: Correlation between endothelial protein C receptor (EPCR) expression level and animal survival.

**Source data 4.** Raw data for *Figure 1F*: Donor chimerism in peripheral blood (PB) myeloid cells 16 weeks post-transplantation.

**Figure supplement 1.** The peripheral blood (PB) lineage output from EPCR$^{high}$CD41$^-$ and EPCR$^{high}$CD41$^+$ cells after transplantation.

**Figure supplement 1—source data 1.** Raw data for *Figure 1—figure supplement 1A*: Donor chimerism in peripheral blood (PB) B and T cells 16 weeks post-transplantation.

**Figure supplement 1—source data 2.** Raw data for *Figure 1—figure supplement 1C*: Cellularity of whole culture expanded ex vivo from EPCR$^{low}$ or EPCR$^{high}$ SLAM LSKs.

Next, single EPCR$^+$ SLAM LSKs were index-sorted (*Figure 1—figure supplement 1B*) and expanded ex vivo in an F12-polyvinyl alcohol (PVA)-based culture media (*Wilkinson et al., 2020b*). At an early stage (8 days, *Figure 1—figure supplement 1C*), cultures initiated with SLAM LSK cells with higher EPCR expression proliferated less compared to those with lower EPCR expression. However, after longer expansion (25 days, *Figure 1—figure supplement 1C*), the EPCR higher cells generated on average a larger amount of progeny. To evaluate the HSC activity following ex vivo expansion and to correlate this directly to EPCR expression levels, the expanded cells from index-sorted cultures of cHSCs were collected and transplanted into individual lethally irradiated mice (*Figure 1—figure supplement 1D*). While freshly and stringently isolated HSCs normally fail to rescue mice from lethal irradiation (*Nakauchi et al., 1999*), we here entertained that culturing of HSCs in addition to inducing self-renewal would also generate progenitors that could radioprotect the hosts. If so, survival could be used as a proxy for combined hematopoietic stem and progenitor cell (HSPC) activity. Indeed, mice receiving progeny from SLAM LSK EPCR higher cells were more efficiently radioprotected (*Figure 1E*). Thereafter, we initiated cultures with EPCR$^{high}$ or EPCR$^{lower}$ SLAM LSK cells fractions (*Figure 1—figure supplement 1E*). Mice transplanted with cells expanded from EPCR$^{high}$ cells radioprotected 5/5 recipients, while only two out of five animals receiving the progeny of EPCR$^{low}$ cells survived long term. As expected, these surviving mice displayed exclusive test cell-derived myelopoiesis long term after transplantation (*Figure 1F*).

Taken together, these experiments established that HSC activity was restricted to EPCR$^{high}$ HSCs, and such cells could effectively generate progeny in vitro that rescue mice from lethal irradiation. Therefore, we operationally defined our input cHSCs by their EPCR$^{high}$ SLAM phenotype.

## Phenotypic heterogeneity from ex vivo expanded cHSCs

The PVA-dependent cell culture system was reported to efficiently support HSC activity ex vivo over several weeks (*Wilkinson et al., 2019*). However, the functional HSC frequency did not appear on par with the total amount of cells generated by the end of the culture period, suggesting the generation also of many differentiated cells (*Wilkinson et al., 2019*). To detail this, we expanded aliquots of cHSCs ex vivo for 14–21 days, followed by multicolor phenotyping. From 50 cHSCs, an average of 0.5 million (14 days) and 13.6 million cells (21 days) were generated (*Figure 2—figure supplement 1A*) and which associated with a phenotypic heterogeneity that increased over time (*Figure 2—figure supplement 1B*). With the assumption that HSCs might retain their cell surface phenotype in cultures, we quantified EPCR$^{high}$ SLAM cells and observed a decrease in their frequency over time, although this was numerically counteracted by the overall proliferation in cultures (*Figure 2A*). Thus, quantification of cHSCs translated into an average expansion of 54-fold (26- to 85-fold) and 291-fold (86- to 1273-fold) at 14 and 21 days, respectively (*Figure 2A*).

To gain a deeper understanding of the cells generated in our ex vivo cultures, we collected cells from the entire culture expanded from 500 cHSCs and isolated nuclei for single-cell multiome sequencing using a commercial platform from 10x Genomics. This approach combines single-cell RNA sequencing and single-cell ATAC sequencing, allowing for integrated analysis of these modalities.

Based on the transcriptomic signatures and distal motif identities, we were able to identify not only early HSPCs, but also many differentiated myeloid cells in the expansion cultures (*Figure 2B*), however with little evidence for lymphoid differentiation. Assessment of a condensed HSC signatures (*Figure 2—figure supplement 1C*) confirmed that the cHSCs were primarily located to cluster 5

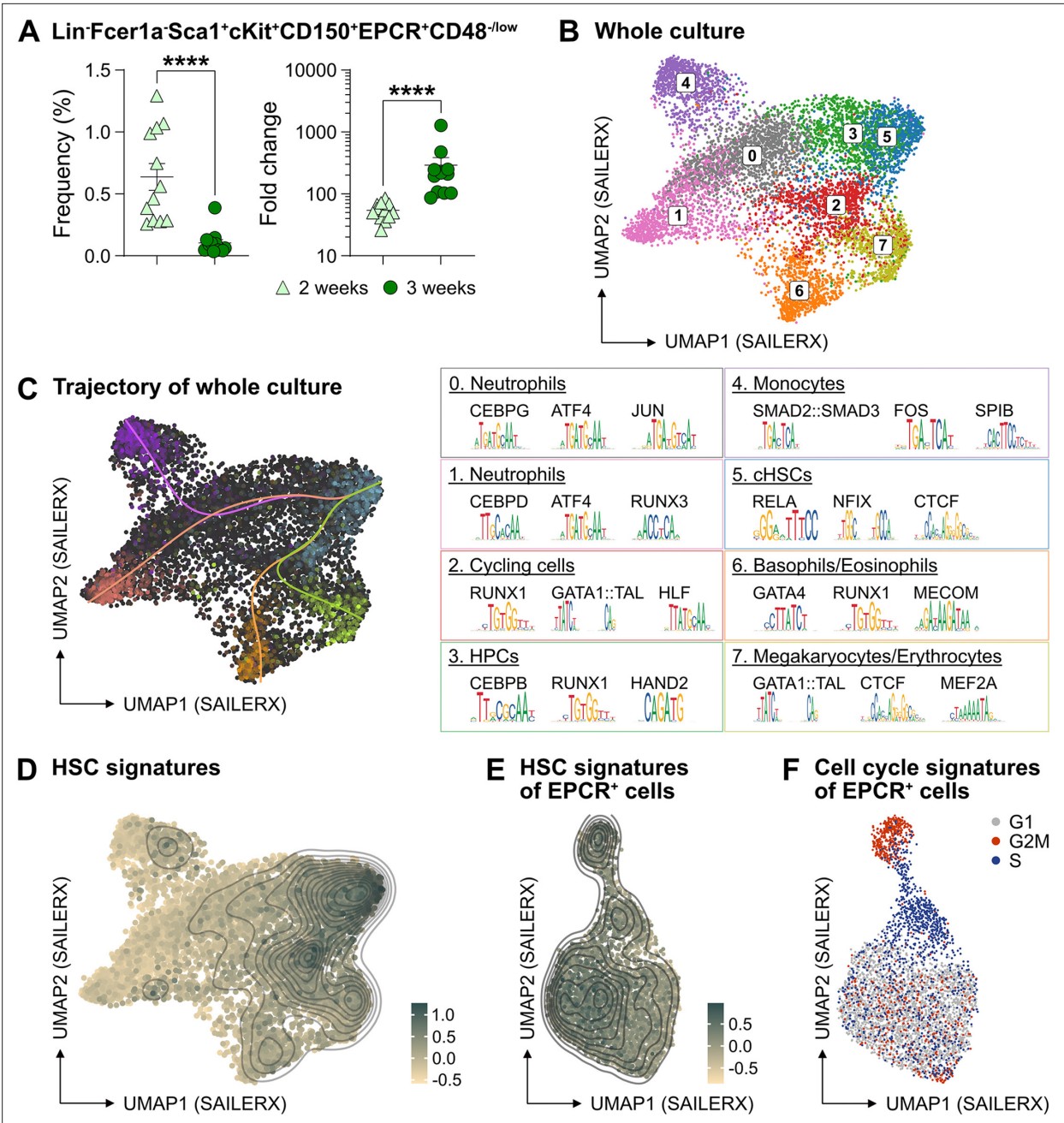

**Figure 2.** The phenotypic properties of hematopoietic stem cell (HSC) expansion cultures. (**A**) Frequency and fold change of phenotypic candidate HSCs (cHSCs) (EPCR[high] SLAM LSKs) in ex vivo cultures after 14 or 21 days of culture (n=12 per group). Data points depict values from individual cultures initiated from 50 cHSCs. Error bars denote SEM. The asterisks indicate significant differences. ****, p<0.0001. (**B**) UMAP (based on SAILERX dimensionality reduction) of single-cell multiome profiling of cells expanded ex vivo for 21 days. Cell-type annotations were derived using marker gene signatures and distal motif identities. (**C**) Trajectory analysis of lineage differentiation for cells expanded ex vivo (left), with the top 3 scoring TF motifs of each cluster (right). (**D**) Expression of HSC signature on whole culture. (**E**) Expression of HSC signature of EPCR[+] cells sorted from ex vivo cultures. (**F**) Cell cycle phase classification of EPCR[+] cells sorted from ex vivo cultures.

The online version of this article includes the following source data and figure supplement(s) for figure 2:

**Source data 1.** Raw data for *Figure 2A*: Frequency and fold change of phenotypic candidate hematopoietic stem cells (cHSCs) expanded ex vivo.

**Figure supplement 1.** Heterogeneity of ex vivo expanded candidate hematopoietic stem cells (cHSCs).

**Figure supplement 1—source data 1.** Raw data for *Figure 2—figure supplement 1A*: Cellularity of whole culture expanded ex vivo for 2 or 3 weeks.

**Figure supplement 1—source data 2.** Raw data for *Figure 2—figure supplement 1B*: Frequency of cells with different surface marker expression patterns.

(*Figure 2D*), with trajectory analyses suggesting multiple differentiation pathways from early HSPCs to differentiated myeloid progeny (*Figure 2C*).

Since the EPCR expression also condensed to cluster 5 (*Figure 2—figure supplement 1D*), we further investigated whether the expression of EPCR marks cHSCs ex vivo by collecting ex vivo expanded EPCR+ cells for single-cell multiome sequencing. These cells displayed less heterogeneity than what we observed for the total culture (*Figure 2B*), which included a more even distribution of the condensed HSC signature (*Figure 2E*) as well as the expression of EPCR (*Figure 2—figure supplement 1E*). Here, the most pronounced separator for the interrogated cells attributed to the cell cycle position (*Figure 2F*).

To explore if other markers could provide additional phenotypic information for genuine HSCs in cultures, we applied an *Fgd5* reporter mouse model previously shown to selectively mark HSCs in situ (*Gazit et al., 2014*). While the *Fgd5* reporter only marked a subset of cells in culture (*Figure 2—figure supplement 1F*), we noted heterogeneous expression of CD48 within the Fgd5 positive fraction (*Figure 2—figure supplement 1F–G*) and that we failed to clearly demarcate in our single-cell multimodal data.

Collectively, these results demonstrated robust expansion of cHSCs using the PVA-based culture system, albeit with additional parallel generation of a large number of more differentiated cells. While the identities of most of the cells in cultures could be clearly deduced by their combined gene expression and chromatin accessibility profiles, this approach failed to inform on the heterogeneous expression pattern of CD48 on cHSCs (*Figure 2—figure supplement 1*).

## Functional HSC activity associates to a minor LSK SLAM EPCR^high fraction within ex vivo cultures

Given our inability to distinctly identify cHSCs by their molecular profiles, we next explored the possibility to prospectively isolate HSCs from the cHSC cultures and instead rely on their functional ability to long-term repopulate lethally irradiated hosts for readout. cHSCs were isolated and cultured ex vivo for 21 days and the expanded cells were separated into two equal portions: one portion was kept unfractionated, while three subpopulations of CD150+LSKs: EPCR^high CD48^-/low, EPCR^high CD48^+, and EPCR^- were sorted from the other portion. The fractions were competitively transplanted into lethally irradiated mice, such that each recipient received fractions equivalent to the expansion from 50 cHSCs (EE50) along with 500,000 competitor WBM cells (*Figure 3A*).

Stable and very high long-term multilineage reconstitution was observed in all recipients of unfractionated cultured cells. By contrast, test cell-derived reconstitution was not recovered from EPCR^- cells (*Figure 3B* and *Figure 3—figure supplement 1*). EPCR^high CD48^+ cells produced very high reconstitution levels short term after transplantation, but the chimerism provided by these cells dropped considerably over time (*Figure 3B* and *Figure 3—figure supplement 1*). This differed strikingly from recipients receiving the minor fraction of EPCR^high CD48^-/low cells, which presented with robust long-term multilineage reconstitution (*Figure 3B* and *Figure 3—figure supplement 1*). While almost no BM-cHSC chimerism was observed from EPCR^high CD48^+ cells, EPCR^high CD48^-/low cells produced very high chimerism in all evaluated progenitor fractions, including for cHSCs (*Figure 3C*).

These data established that functional in vivo HSC activity following culture is restricted to the minor fraction of SLAM LSK EPCR^high cells.

## Quantification of HSC activity in ex vivo expansion cultures

Expanded cHSCs presented with a vastly higher reconstitution activity compared to freshly isolated cHSCs, and mostly fell out of range for accurate quantification (*Figure 3B*). We therefore assessed the impact of lowering the amount of input cHSCs and enhancing the amount of competitor cells. Aliquots of 10 cHSCs were cultured and expanded for 21 days. Cells from each culture (EE10) were next mixed with 2 or 20 million (2M or 20M) competitor WBM cells, followed by transplantation (*Figure 4A*, 'individual'). In parallel, we also included a group in which EE50 cHSCs were mixed with 10 million WBM cells and split among five lethally irradiated recipients (*Figure 4A*, 'pooled'). This allowed us to assess the variability among input cHSCs.

Although robust reconstitution was observed in most recipients of 'individual' cultures (*Table 1*, *Figure 4B–C*, and *Figure 4—figure supplement 1A–B*), the levels of test cell-derived cells varied extensively among these recipients. To quantify the HSC activity in each recipient, we calculated

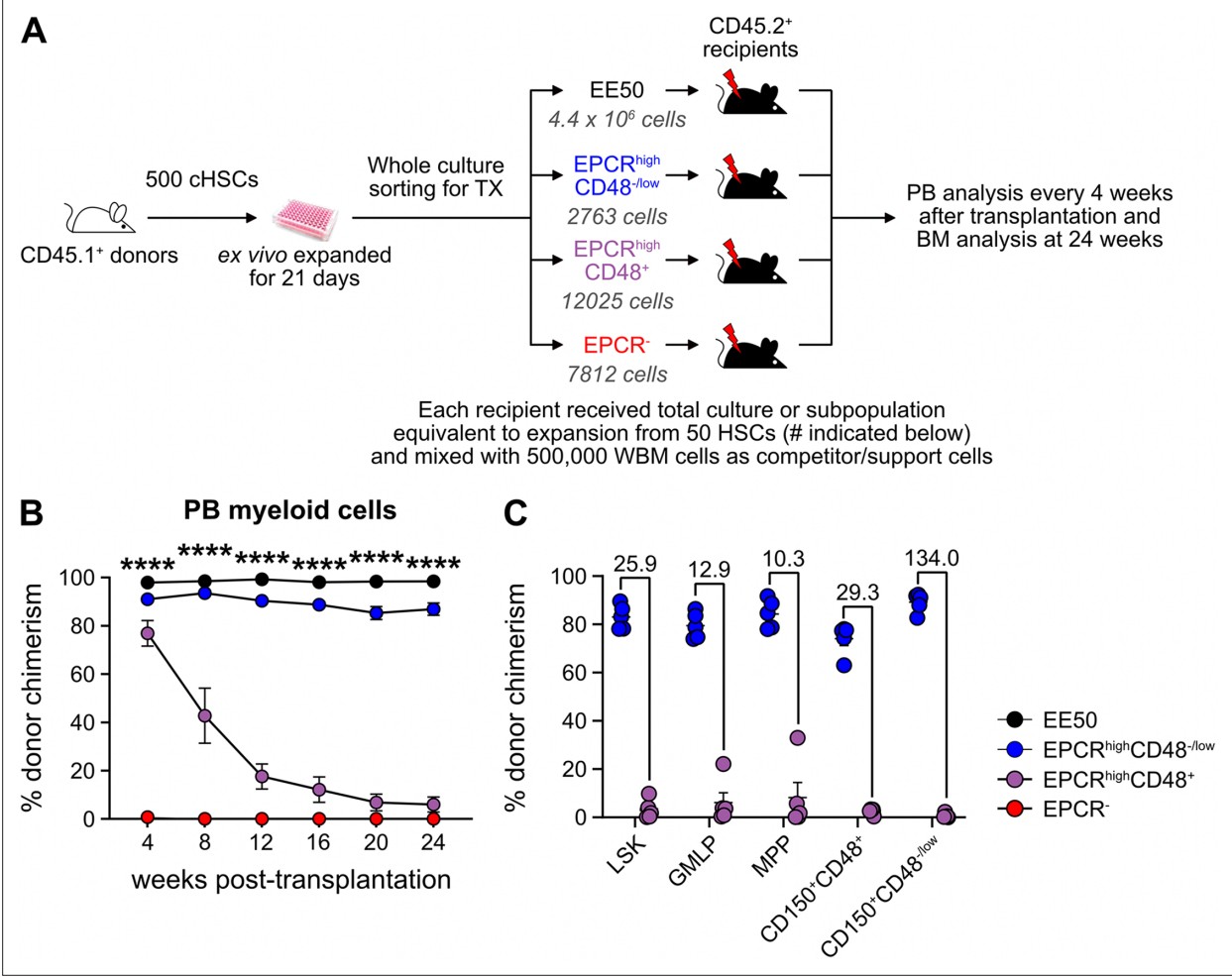

**Figure 3.** Hematopoietic stem cell (HSC) activity can be prospectively isolated from candidate HSC (cHSC) cultures and associates to a minor EPCR⁺ SLAM LSK fraction. (**A**) Strategy to assess the in vivo HSC activity of subfractions from ex vivo cultures. (**B**) Test cell-derived chimerism in peripheral blood (PB) myeloid lineages over 24 weeks post-transplantation. Data represent mean values ± SEM (n=5 per group). A one-way ANOVA test was applied, with the asterisks indicating significant differences among the four groups. ****, p<0.0001. (**C**) Test cell-derived chimerism in bone marrow (BM) progenitor subsets 24 weeks post-transplantation (n=5 per group). Numbers indicate fold differences between the EPCR⁺ CD48⁻/low and EPCR⁺CD48⁺ fractions, and data points depict chimerism levels in individual recipients.

The online version of this article includes the following source data and figure supplement(s) for figure 3:

**Source data 1.** Raw data for *Figure 3B*: Donor chimerism in peripheral blood (PB) myeloid cells over 24 weeks post-transplantation.

**Source data 2.** Raw data for *Figure 3C*: Donor chimerism in bone marrow (BM) progenitors 24 weeks post-transplantation.

**Figure supplement 1.** B and T cell chimerism in peripheral blood (PB) after transplantation.

**Figure supplement 1—source data 1.** Raw data for *Figure 3—figure supplement 1A*: Donor chimerism in peripheral blood (PB) B cells over 24 weeks post-transplantation.

**Figure supplement 1—source data 2.** Raw data for *Figure 3—figure supplement 1B*: Donor chimerism in peripheral blood (PB) T cells over 24 weeks post-transplantation.

the repopulating units (RUs) (*Harrison et al., 1993*) for each lineage and recipient separately at the experimental endpoint (*Table 1*). Short-lived myeloid cells have been proposed as a better indicator of ongoing HSC activity (*Domen et al., 2000*), and in agreement with this, we observed a high correlation between the RU^myeloid and RUs for cHSCs in the BM (*Table 1*). A much more consistent HSC activity was observed in-between recipients of 'pooled' cells (*Table 1*, *Figure 4B–C*). This confirmed that the differential chimerism in recipients of limited cHSC numbers relates to an unresolved cellular heterogeneity of input cHSCs. RUs per initial cHSC were calculated for freshly isolated BM-cHSCs (the three EPCR^high groups from *Figure 1B*) and following 3-week culture ('pooled' group). This revealed

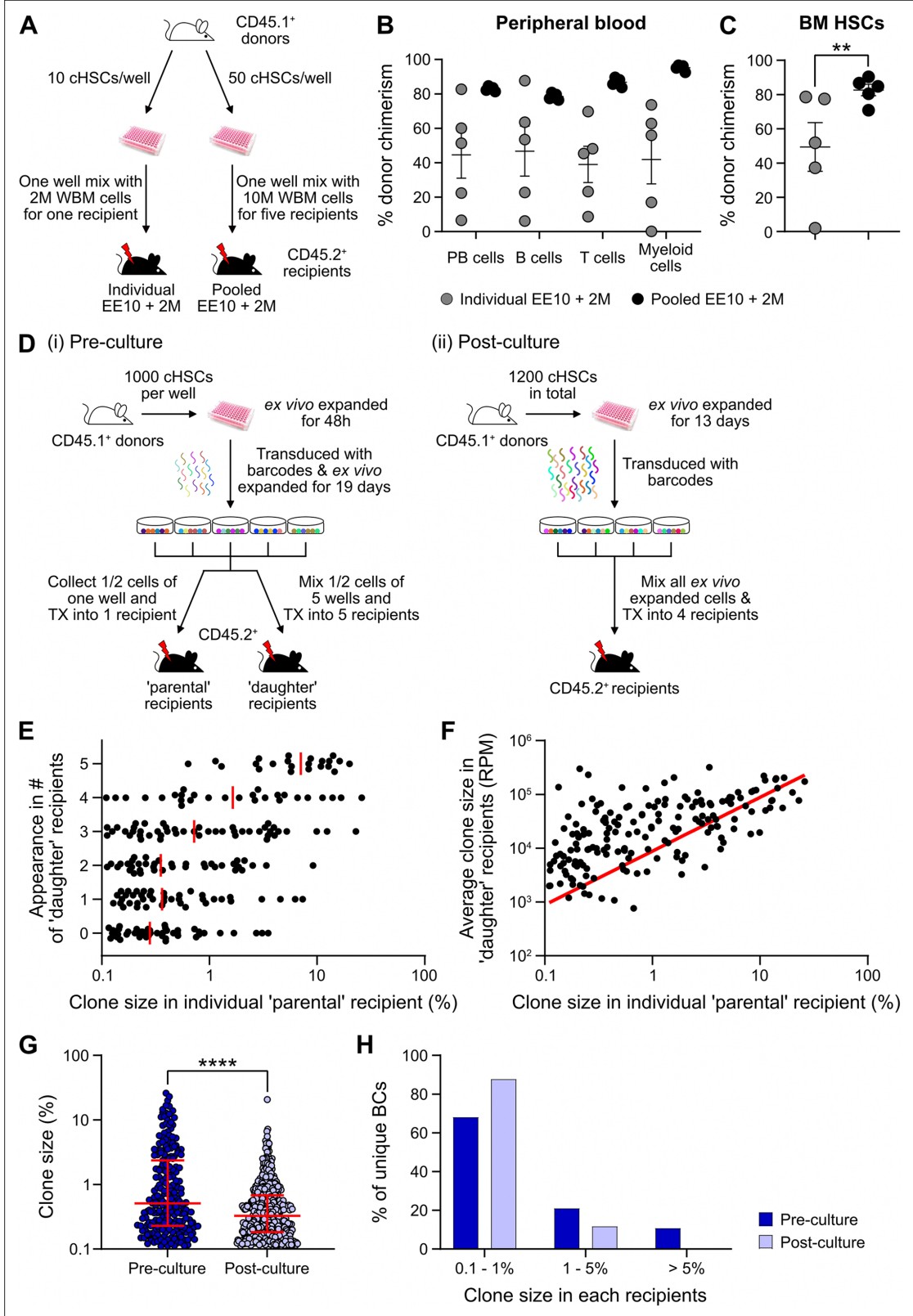

**Figure 4.** Quantification of repopulating activity from expanded candidate hematopoietic stem cells (cHSCs). (**A**) Competitive transplantation strategies used to assess the repopulation of ex vivo expanded cHSCs. (**B**) Test cell-derived peripheral blood (PB) reconstitution 16 weeks post-transplantation (n=5 per group). Symbols denote individual mice and means ± SEM. (**C**) Bone marrow (BM) cHSCs chimerism 16 weeks post-transplantation (n=5 per group). Symbols denote individual mice, and means ± SEM. (**D**) Barcode approaches used to assess the clonal HSC contribution of ex vivo expanded

*Figure 4 continued on next page*

*Figure 4 continued*

HSCs before (**i**) or after (**ii**) ex vivo expansion. (**E**) Clone sizes of unique barcodes in 'parental' recipients and their appearance in 'daughter' recipients, demonstrating extensive variation in expansion capacity among individual HSCs. Red lines indicate the median clone size in 'parental' recipients. (**F**) Clone sizes of unique barcodes detected in BM myeloid cells of 'parental' recipients and their corresponding contribution in 'daughter' recipients. The red line denotes the correlation/linear regression. (**G**) Clone sizes in 'parental' recipients transplanted with 'pre-culture' barcoded cells, or in recipients of 'post-culture' barcoded cells. Median clone sizes are shown with interquartile ranges. (**H**) Frequency of barcodes and their clone sizes in recipients of pre- or post-cultured barcoded HSCs. BCs, barcodes. All data points depict values in individual recipients or barcodes. Asterisks indicate significant differences. *, p<0.05; ****, p<0.0001.

The online version of this article includes the following source data and figure supplement(s) for figure 4:

**Source data 1.** Raw data for *Figure 4B*: Donor chimerism in peripheral blood (PB) lineages 16 weeks post-transplantation.

**Source data 2.** Raw data for *Figure 4C*: Donor chimerism in bone marrow (BM) hematopoietic stem cells (HSCs) 16 weeks post-transplantation.

**Source data 3.** Raw data for *Figure 4E*: Clone sizes of unique barcodes in 'parental' recipients and their appearance in 'daughter' recipients.

**Source data 4.** Raw data for *Figure 4F*: Clone sizes of unique barcodes detected in bone marrow (BM) myeloid cells of 'parental' recipients and their corresponding contribution in 'daughter' recipients.

**Source data 5.** Raw data for *Figure 4G*: Clone size distribution of pre- and post-culture barcodes.

**Source data 6.** Raw data for *Figure 4H*: Frequency distribution of pre- and post-culture barcodes.

**Figure supplement 1.** Quantification of hematopoietic stem cell (HSC) activity from cultured bone marrow (BM) or fetal liver (FL)-HSCs.

**Figure supplement 1—source data 1.** Raw data for *Figure 4—figure supplement 1A*: Donor chimerism in peripheral blood (PB) lineages 16 weeks post-transplantation.

**Figure supplement 1—source data 2.** Raw data for *Figure 4—figure supplement 1B*: Donor chimerism in bone marrow (BM) hematopoietic stem cells (HSCs) 16 weeks post-transplantation.

**Figure supplement 1—source data 3.** Raw data for *Figure 4—figure supplement 1D*: Whole culture cellularity and frequency and fold change of phenotypic candidate hematopoietic stem cells (cHSCs) expanded ex vivo from bone marrow (BM)- or fetal liver (FL)-HSCs.

**Figure supplement 1—source data 4.** Raw data for *Figure 4—figure supplement 1E*: Donor chimerism in peripheral blood (PB) lineages 16 weeks post-transplantation.

**Figure supplement 1—source data 5.** Raw data for *Figure 4—figure supplement 1F*: Donor chimerism in bone marrow (BM) hematopoietic stem cells (HSCs) 16 weeks post-transplantation.

**Figure supplement 1—source data 6.** Raw data for *Figure 4—figure supplement 1G*: Repopulating units (RUs) per initial hematopoietic stem cells (HSCs) in peripheral blood (PB) myeloid cells and bone marrow (BM) HSCs.

**Table 1.** Repopulating units (RUs) for each lineage in peripheral blood (PB) and for bone marrow (BM) candidate hematopoietic stem cells (cHSCs) of each recipient.

| Recipients | | PB | B cells | T cells | Myeloid cells | BM-cHSCs |
|---|---|---|---|---|---|---|
| | #1 | 6 | 6 | 6 | 4 | 12 |
| | #2 | 31 | 35 | 19 | 27 | 75 |
| | #3 | 21 | 23 | 17 | 35 | 70 |
| | #4 | 98 | 144 | 46 | 57 | 21 |
| *Individual BM EE10 +2M* | #5 | 1 | 1 | 2 | 0.03 | 0.4 |
| | #1 | 101 | 78 | 173 | 583 | 49 |
| | #2 | 110 | 84 | 120 | 516 | 82 |
| | #3 | 87 | 68 | 103 | 248 | 130 |
| | #4 | 101 | 70 | 149 | 384 | 187 |
| *Pooled BM EE10 +2M* | #5 | 65 | 127 | 186 | 331 | 112 |

1 RU equals to the average reconstitution activity of 1×105 WBM cells.

461- and 70-fold increases in PB myeloid and BM-cHSC chimerism in recipients of ex vivo expanded cells, respectively. This is an estimate of the increase of functional HSC activity following ex vivo culture (*Figure 4B–C* and *Figure 4—figure supplement 1G*).

We next assessed HSC clonality using a lentiviral barcoding approach. Five aliquots of 1000 cHSCs were transduced with a barcode library (*Biddy et al., 2018*) 48 hr after culture and then expanded for 19 days. Half of the expanded cells from each well were transplanted into one 'parental' recipient, respectively, while the other half was mixed with each other, with each 'daughter' recipient receiving 1/5 of the mixed cells (*Figure 4D-i*). As a proxy for the HSC activity, barcodes were extracted from BM test-derived myeloid cells 16 weeks post-transplantation. Following stringent filtering, we retrieved 223 unique barcodes in the 'parental' recipients, with a highly variable contribution (0.1–26.0%, *Figure 4E–G*). Representation of the same barcode in 'parental' and 'daughter' recipients demonstrates a shared clonal/HSC origin, where more robustly expanding HSCs should have a higher chance of reconstituting more 'daughter' recipients. Our data agreed well with this presumption (*Figure 4E*), and the most actively shared clones also associated with larger outputs in 'daughter' recipients (*Figure 4F*).

While our initial barcode experiments were designed to assess the in vitro expansion potential of cHSCs, a next set of experiments were executed to assess the clonal HSC activity following expansion. For this, cHSCs were ex vivo expanded for 13 days, provided with barcodes (*Horlbeck et al., 2016*) overnight, and transplanted into four lethally irradiated recipients (*Figure 4D-ii*). 16 weeks post-transplantation, we recovered 573 unique barcodes. The median clone sizes of HSCs post-culture were significantly smaller than the pre-culture clones (*Figure 4G*), which was expected as more clones (573 vs 223) were evaluated in the 'post-culture' setting. However, the frequency of dominantly contributing clones was significantly less abundant (*Figure 4H*), demonstrating a more even contribution from individual clones in this setting. This is in line with the interpretation that expanded HSCs are functionally more equivalent than input cHSCs.

Finally, we assessed how the culture system supports the in vivo activity of fetal liver (FL)-cHSCs (*Figure 4—figure supplement 1C*). E14.5 FL-cHSCs were sorted using the same immuno-phenotype as BM-cHSCs and cultured ex vivo. We observed an average 188-fold increase of cHSCs following 21 days' culture (*Figure 4—figure supplement 1D*). We used the 'pooled' approach for functional evaluation and competed EE10 cells with either 2 or 20 million WBM cells (*Figure 4—figure supplement 1C and E*). While ex vivo expanded FL-cHSCs contributed considerably to multilineage reconstitution (*Figure 4—figure supplement 1E*), we observed 5.7- and 6.6-fold lower reconstitution within PB myeloid cells and cHSCs when comparing to animals transplanted cells expanded from BM-cHSCs, respectively (*Figure 4—figure supplement 1F–G*).

Taken together, these experiments demonstrated robust increases in HSC activity following culture of cHSCs, albeit slightly less for FL-cHSCs. However, clonal barcode assessments revealed substantial variation in ex vivo expansion potential of even stringently purified input cHSCs.

## Ex vivo expanded cHSCs returns to a quiescent state following engraftment in unconditioned recipients

Successful engraftment in an unconditioned setting requires large numbers of HSCs/HSPCs (*Brecher et al., 1982*), which reportedly can be accommodated by PVA-cultured HSCs (*Wilkinson et al., 2019*). To test this, we transplanted EE100 CD45.1+ cHSCs into lethally irradiated or unconditioned CD45.2+ recipients (*Figure 5—figure supplement 1A*). In lethally irradiated hosts, cultured cells reconstituted >90% of PB cells, including with robust contribution of donor cells to the myeloid lineages. By contrast, most of the unconditioned recipients lacked long-term reconstitution (*Figure 5—figure supplement 1B*). Given that CD45 mismatching might be a sufficient immunological barrier to prevent HSPC reconstitution (*van Os et al., 2001*), we instead transplanted expanded cells into unconditioned F1 CD45.1+/CD45.2+ hosts (*Figure 5A*). Clear long-term multilineage engraftment was observed in all these hosts (*Figure 5B*).

In the BM, cHSCs exist for the most part in a quiescent state, which contrasts the situation in cultures (*Figure 5—figure supplement 1*). To investigate to what extent ex vivo expanded cHSCs could return to quiescence following transplantation, we expanded 100 CD45.2+ cHSCs for 21 days and stained all their progeny with the proliferation-tracking dye CellTrace Violet (CTV) prior to transplantation into CD45.1+/CD45.2+ hosts. As a control, we transplanted CTV-labeled unmanipulated CD45.2+

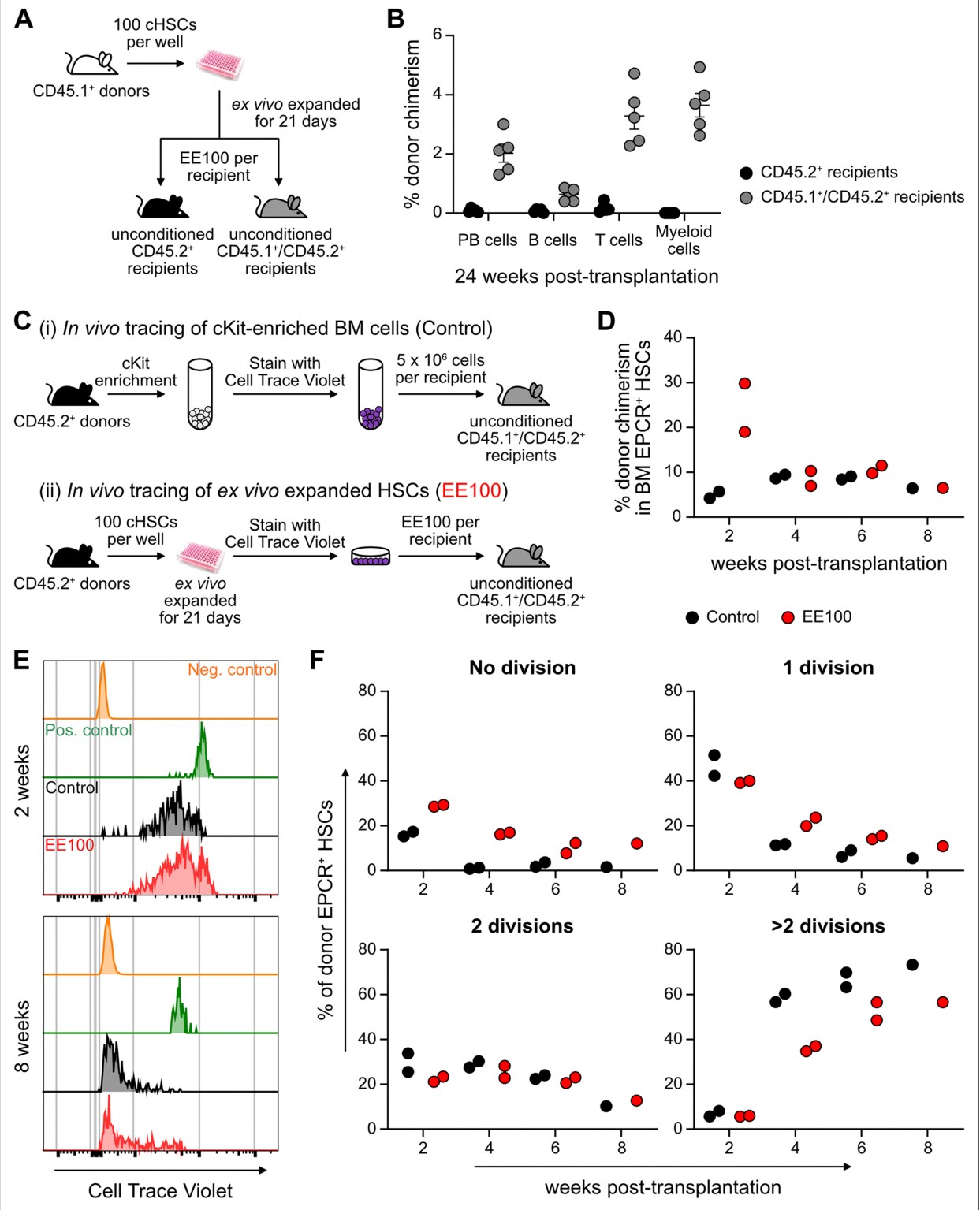

**Figure 5.** Cultured candidate hematopoietic stem cells (cHSCs) allow for transplantation into non-conditioned syngeneic recipients. (**A**) Strategy to assess the ability of ex vivo expanded cHSCs to engraft unconditioned recipients. (**B**) Test cell-derived peripheral blood (PB) reconstitution 24 weeks post-transplantation (n=5 per group). Symbols denote individual mice and means ± SEM. (**C**) Strategy used to assess the in vivo proliferation of ex vivo expanded cHSCs. (**D**) Bone marrow (BM) cHSC chimerism 2–8 weeks post-transplantation. n=2 per group for 2, 4 or 6 weeks post-transplantation. n=1 per group for 8 weeks post-transplantation. (**E**) Representative CellTrace Violet (CTV) label profiles of donor EPCR+ cHSCs compared to negative control signal (host EPCR+ cHSCs) and positive signal (donor CD4+ spleen cells) at 2 or 8 weeks post-transplantation. (**F**) Donor EPCR+ cHSCs were evaluated for

*Figure 5 continued on next page*

*Figure 5 continued*

the number of cell divisions they had undergone through 8 weeks post-transplantation. n=2 per group for 2, 4 or 6 weeks post-transplantation. n=1 per group for 8 weeks post-transplantation. All data points depict values in individual recipients.

The online version of this article includes the following source data and figure supplement(s) for figure 5:

**Source data 1.** Raw data for *Figure 5B*: Donor chimerism in peripheral blood (PB) lineages 24 weeks post-transplantation.

**Source data 2.** Raw data for *Figure 5D*: Donor chimerism in bone marrow (BM) EPCR$^+$ hematopoietic stem cells (HSCs) over 8 weeks post-transplantation.

**Source data 3.** Raw data for *Figure 5F*: Cell divisions of donor bone marrow (BM) EPCR$^+$ hematopoietic stem cells (HSCs).

**Figure supplement 1.** The fate of ex vivo cultured candidate hematopoietic stem cells (cHSCs) following transplantation into unconditioned hosts.

**Figure supplement 1—source data 1.** Raw data for *Figure 5—figure supplement 1B*: Donor chimerism in peripheral blood (PB) lineages 16 weeks post-transplantation.

**Figure supplement 1—source data 2.** Raw data for *Figure 5—figure supplement 1G* Number of undivided donor EPCR$^{high}$ hematopoietic stem cells (HSCs) without division.

cKit-enriched BM cells (*Figure 5C*). In these animals, splenic CD4$^+$-enriched cells were co-transplanted as a control for non-dividing cells (*Figure 5—figure supplement 1D*). Reconstitution and cHSC fates were monitored biweekly for up to 8 weeks post-transplantation. Although the phenotypic cHSC contribution in recipients receiving ex vivo expanded cells outnumbered the contribution from unmanipulated cells 2 weeks post-transplantation, the levels became comparable later (*Figure 5D*). Recipients from both groups showed high levels of CTV signal 2 weeks post-transplantation, with the majority of donor EPCR$^+$ cHSCs having undergone fewer than three cell divisions. Over time, increasing numbers of cHSCs had proliferated within both groups and with similar patterns (*Figure 5E–F* and *Figure 5—figure supplement 1E*), with a slightly higher number of undivided cHSCs retrieved from ex vivo expanded HSCs at all time points (*Figure 5E*). The most quiescent cHSCs associated with higher EPCR levels (*Figure 5—figure supplement 1F–G*), again attesting to the usefulness of EPCR to predict robust HSC activity.

Therefore, ex vivo expanded cHSCs can efficiently engraft immunologically matched hosts without conditioning. Under such conditions, expanded cHSCs can rapidly return to quiescence following transplantation.

## Discussion

Although many efforts have aimed to in vitro expand murine HSCs, the results have for the most part been disappointing (*Wilkinson et al., 2020a*). A recent study suggested that the substitution of undefined serum products with the synthetic polymer PVA is one key determinant toward this goal (*Wilkinson et al., 2019*). Here, we detailed this culture system further. We focused on the requirements of candidate input HSCs, the reproducibility of the system, the cellular output, and the functional in vivo performance of the in vitro replicating HSCs. Several observations that we believe are valuable not only for future applications of the PVA culture system, but also for HSC biology in general, emerged from our work.

While the PVA system supports HSC activity even from unfractionated BM cell preparations (*Ochi et al., 2021*), the long-term repopulating HSCs (LT-HSCs) in cruder input cell preparations are very low. While having its own advantages, this makes it difficult to track the fates of cHSCs. Therefore, we here consistently initiated cultures with more stringently purified cHSCs. As a starting point, we used cHSCs isolated based on a current standard SLAM phenotype (Lin$^-$Sca$^+$cKit$^+$CD48$^-$CD150$^+$) (*Challen et al., 2021*). In agreement with previous reports (*Kent et al., 2009*), we observed that CD201/EPCR further enriches for LT-HSCs, a knowledge that has so far not been adopted as a standard in the field (*Challen et al., 2021*). This applied both to cHSCs isolated directly from the BM and following culture. In fact, despite that HSCs change many phenotypes in culture (*Zhang and Lodish, 2005*), the LT-HSC activity following culture was remarkably enriched not only in the EPCR positive fraction, but also for other phenotypic attributes of unmanipulated LT-HSCs. Importantly, such cHSCs represented only a very minor fraction in cultures (0.6% and 0.1% of 2- and 3 week cultures, respectively) but contained all LT-HSC activity, with the massive overall expansion in cultures leading to substantial net expansions of such cells (100- to 1200-fold).

While we gained insights into the dynamics of ex vivo differentiation of cHSCs into mature lineages by simultaneously measuring ATAC signal and RNA transcripts in individual nuclei using single-cell multiome sequencing, the limited sensitivity of this method hindered the identification of more homogeneous populations of bona fide HSCs within the sorted EPCR⁺ cells. Therefore, we assessed the functional in vivo HSC activity in most of our subsequent work. While our clonal barcoding experiments unequivocally demonstrated HSC self-renewal, the large spectra in expansion potential of individual cHSCs was noteworthy. This variation was reduced when assessing HSC activity following culture and aligns with the interpretation that the offspring of cHSCs that self-renew robustly in cultures exhibit less differences. Nonetheless, the demonstrated in vitro self-renewal refutes alternative views in which the culture process would mainly act to enhance the reconstitution capacity of individual HSCs, for instance by altering their homing properties and/or by the conversion of more differentiated progenitors into cells with LT-HSC activity. Therefore, our data support the view that this (or other) culture systems might not support all types of LT-HSCs previously alluded to *Haas et al., 2018*. Further support to this notion can be derived from our data on FL-cHSCs, which did not expand better than adult HSCs in the PVA system, despite that FL-HSCs repopulate mice more efficiently than adult HSCs when unmanipulated (*Rebel et al., 1996*). In a broader sense, this associates also with the general concept of self-renewal and that has recently been highlighted in studies on native hematopoiesis, where distinct but only slowly contributing LT-HSCs co-exist with more active progenitors that can presumably also execute self-renewal divisions (*Sun et al., 2014*; *Busch et al., 2015*; *Säwen et al., 2018*). It appears that many aspects of this structure are re-created in cultures and, as we show, this requires as input the normally very slowly dividing and presumably most primitive LT-HSCs.

While the parallel self-renewal and differentiation in HSC cultures represent a fundamental difference compared to other in vitro stem cell self-renewal systems (e.g. for ES/iPS cells), where preferential self-renewal can be achieved, the differentiation of HSCs can be taken advantage of. We observed that even single cultured HSCs can enhance the speed of donor reconstitution and alleviate the effects of myeloablative conditioning, and the parallel self-renewal and differentiation can also be harnessed to approach HSC fate decisions at a molecular level (*Weinreb et al., 2020*). This was exemplified by our single-cell multimodal experiments, which allowed for determinations of the differentiation trajectories of cHSCs in the culture system and their associated molecular features. However, as the expansion of genuine LT-HSC activity is far from unlimited, a shortage of HSCs is by all likelihood still an obstacle for larger-scale screening efforts such as genome-wide CRISPR screens and larger small molecule screens.

During our studies, we encountered several aspects of the CRA that are often neglected. First, the HSC activity when transplanted at limited numbers highlighted large variations in between recipients, which is in line with previous extensive single-cell transplantation experiments (*Yamamoto et al., 2018*; *Yamamoto et al., 2013*; *Benz et al., 2012*; *Carrelha et al., 2018*; *Oguro et al., 2013*). While raised previously (*Ema and Nakauchi, 2000*), taking into account not only the repopulating activity but also clonal aspects has not reached wide use. Emerging barcoding technologies, by which many clones can be evaluated in a single mouse, makes this easier. Second, the progeny from a relatively small number of cultured cHSCs (10–50 cells) vastly outcompeted 'standard' doses of competitor cells, making quantification unreliable/impossible. This contrasted results from freshly isolated cHSCs transplanted at the same doses used to initiate cultures. While this concern could be overcome by adjusting the graft composition (e.g. enhancing the number of competitor cells), extensive variation still existed among cultures that could be eliminated by using larger amounts of input cells and splitting the contents into individual recipients. This should have relevance also for normal HSC biology by highlighting intrinsic rather than stochastic regulation as the central determinant for persistent HSC function. A third consideration relates to the definition of ongoing HSC activity. We and others have previously established that this is best mirrored by the HSC contribution to myelopoiesis (*Norddahl et al., 2011*; *Domen and Weissman, 2003*). When we assessed cultured EPCR⁺CD48⁺ cells (with predominantly transient multilineage activity), we observed prominent long-term T cell contribution but limited myeloid reconstitution. This should not be confused with the concept of HSC lineage-bias, but rather reflects that the pool of T cells in vivo can be maintained by homeostatic proliferation and thus do not require continuous input from HSCs. Thus, short-lived non-self-renewing multipotent progenitors, which are generated in large numbers in culture, can contribute extensive (lymphoid) offspring in the long term but should not be mistaken for ongoing HSC activity. This consideration

is valid also for competitor grafts in the CRA (in the form of WBM cells), which contain numerous progenitor cells that can rapidly and persistently contribute to lymphopoiesis. Thus, as opposed to the classic CRA, where the overall donor contribution is assessed (*Harrison et al., 1993*), the contribution to short-lived lineages reflects better the ongoing HSC activity, with the contribution to lymphoid lineages serving as a qualitative parameter for multipotency. In agreement with previous viral barcoding studies on HSCs (*Lu et al., 2011*), we observed that when cultured HSCs are barcoded and assessed long term, there is a strong correlation between barcodes in BM progenitors and myeloid cells, while the pool of mature B cells contain clones that cannot be recovered in other lineages (data not shown).

Successful repopulation of non-conditioned hosts is based on the suggestion that large numbers of HSCs are needed to saturate those few BM niches available for engraftment (*Bhattacharya et al., 2006*; *Czechowicz et al., 2007*). Apart from evident clinical implications, avoiding toxic myeloablation has also been a precedence in experimental murine HSC biology, where lethal myeloablation enforces vigorous HSC proliferation following transplantation (*Säwén et al., 2016*), but which contrasts native contexts (*Säwén et al., 2016*; *Sun et al., 2014*; *Busch et al., 2015*). We observed that the CD45.1/CD45.2 differences are sufficiently immunogenic to mediate long-term graft failure in the non-conditioned setting, which in hindsight has been alluded to previously (*Bhattacharya et al., 2006*; *van Os et al., 2001*; *Xu et al., 2004*). Thus, by matching hosts for CD45 isoforms, we could successfully obtain low-level (2–5%) long-term multilineage chimerism in adult non-myeloablated hosts from only 100 cHSCs. Intriguingly, despite being activated to proliferate in vitro, many cHSCs rapidly returned to a quiescent state in vivo, with the most dormant cHSCs exhibiting the most stringent HSC phenotype (EPCR$^{high}$ SLAM LSKs). The possibility for non-conditioned transplantation should complement genetic lineage tracing models aimed at exploring both normal and malignant hematopoiesis in more native/physiological contexts (*Säwén et al., 2016*; *Sun et al., 2014*; *Busch et al., 2015*).

In summary, we here characterized and detailed murine HSCs as they self-renew and differentiate in vitro. Although several aspects of the PVA system remain to be explored, a powerful in vitro self-renewal system has been a long-sought goal in HSC biology where HSCs have been notoriously difficult to uphold in an undifferentiated/self-renewal state. The merger of this system with recent advances in transplantation biology, genome engineering, and single-cell techniques holds promise for many exciting discoveries of relevance to both basic and more clinically oriented hematopoietic research.

## Materials and methods

### Mice

Adult (8- to 10-week-old) C57Bl/6N (CD45.2$^+$) female mice were purchased from Taconic. Transgenic *Fgd5*-ZsGreen-2A-CreERT2 mice (*Gazit et al., 2014*) (JAX:027789) was a kind gift from Derrick Rossi. Mice were maintained in the animal facilities at the Biomedical Center of Lund University and kept in environment-enriched conditions with 12 hr light-dark cycles and water and food provided ad libitum. All experimental procedures were approved by a local ethical committee (permits M186-15 and 16468/2020). All efforts were made to reduce the number of experimental animals and suffering.

### BM transplantation

All mice used as recipients were 8–12 weeks of age. For conditioned recipients, mice were lethally irradiated (950 cGy) at least 4 hr prior to transplantation. The conditioned mice received prophylactic ciprofloxacin (HEXAL, 125 mg/l in drinking water) for 2 weeks beginning on the day of irradiation. All transplantations were performed through intravenous tail vein injection. The amount of transplanted test cells and competitor/support cells is described in the Results section.

### In vitro HSC culture

In vitro HSC cultures were performed using F12-PVA-based culture conditions as previously described (*Wilkinson et al., 2019*). In brief, cHSCs (EPCR$^{high}$CD150$^+$CD48$^{-/low}$ LSKs) were sorted into 96-well flat-bottom plates (Thermo Scientific) coated with 100 ng/ml fibronectin (Sigma) for >1 hr at 37°C prior to sorting. HSCs were sorted into 200 µl HSC media (*Supplementary file 1a*) and expanded at 37°C with 5% $CO_2$ for up to 21 days. The first media changes were performed 5 days (for ≥50 initial HSCs), 10 days (for 10 initial HSCs), or 15 days (for single-cell cultures) after initiation of cultures and

then every 2 days. For pre-cultures associated with lentiviral transduction, the first media change was performed 24 hr after transduction and then as above. Cells were split at a 1:2–1:4 ratio into new fibronectin-coated plates when reaching 80–90% confluency (normally every 4 days after the first split). In order to expand cHSCs to reach a workable number as well as avoiding vast numbers of differentiated cells generated after very long term of culture (for example up to 4 weeks), the cHSCs were kept expanding ex vivo for 3 weeks as a standard protocol. Following 14- and/or 21-day expansion, cellularity was assessed using an Automated Cell Counter (TC20, Bio-Rad) and used for flow cytometric analyses and/or transplantation. BM cells collected from wild-type animals were pooled together for sorting to initiate HSC cultures. Each initial well was considered as a technical replicate and treated/analyzed separately.

## Cell preparation

Mice were euthanized by cervical dislocation after which bones (both right and left tibias, femurs, and pelvis) or spleens were extracted. Fetal cells were extracted from livers at E14.5. Bones were crushed using a mortar and pestle and BM cells were collected in ice-cold phosphate-buffered saline (PBS, Gibco) with 1% fetal bovine serum (Sigma-Aldrich) (FACS buffer), filtered (100 µm mesh) and washed. FL or spleen cells were brought into single-cell suspension by grinding through a 100 µm mesh. BM or FL cells were cKit-enriched by anti-cKit-APCeFluor780 (eBioscience) staining and spleen cells were CD4-enrihced by anti-CD4-APC-Cy7 staining, followed by incubation with anti-APC-conjugated beads (Miltenyi Biotec). Magnetic separation was performed using LS columns according to the manufacturer's instructions (Miltenyi Biotec). cKit-enriched BM or FL cells were washed and stained with fluorescently labeled antibodies for analysis or sorting.

PB chimerism analysis after transplantation was done as previously described (*Säwen et al., 2018*). In brief, blood was drawn from tail vein into FACS buffer containing 10 U/ml Heparin (Leo). After incubating with an equal volume of 2% dextran (Sigma) in PBS at 37°C, the upper phase was collected and erythrocytes were lysed using an ammonium chloride solution (STEMCELL Technologies) for 3 min at room temperature, followed by washing.

For analysis and/or sorting after ex vivo expansion, cultured cells were resuspended by pipetting and collected using FACS buffer. An aliquot was taken for cell counting using an Automated Cell Counter (TC20, Bio-Rad).

## Flow cytometry analysis and FACS

Cells were kept on ice when possible, with the FACS buffer kept ice-cold. Staining, analysis, and sorting were performed as previously described (*Säwén et al., 2016*). In brief, cells were resuspended in Fc-block (1:50, $5×10^6$ cells/50 µl, InVivoMab) for 10 min and then for 20 min with a twice concentrated antibody staining mixture (*Supplementary file 1b-j*) at 4°C in dark. In case biotinylated lineage markers were included, a secondary staining with streptavidin BV605 was performed for 20 min (1:400, $5×10^6$ cells/100 µl, Sony) at 4°C in dark. After a final wash, the cells were resuspended in FACS buffer containing propidium iodide (1:1000, Invitrogen).

All FACS experiments were performed at the Lund Stem Cell Center FACS Core Facility (Lund University) on Aria III and Fortessa X20 instruments (BD). Bulk populations were sorted using a 70 µm nozzle, 0.32.0 precision mask, and a flow rate of 2–3K events/s. HSCs for single-cell culture were index-sorted. FACS analysis was performed using FlowJo v10.8.0 (Tree Star Inc).

## Multiome single-cell sequencing

### Library preparation

Multiome sequencing experiments were performed at the Center for Translational Genomics (Lund University) using the Chromium Next GEM Single Cell Multiome ATAC+Gene Expression Reagent Bundle kit according to the manufacturer's instructions (10x Genomics). 40,000 viable cells or $EPCR^+$ viable cells were sorted from ex vivo expanded cultures for multiome single-cell sequencing. Data has been uploaded to GEO under accession number GSE234906.

### Bioinformatic analysis

Data was mainly analyzed with the Seurat (*Hao et al., 2021*) and Signac (*Stuart et al., 2021*; *R Development Core Team, 2022*) packages, as well as the SAILERX Python package (*Cao et al., 2022*)

for the joint-modality dimensionality reduction. All accompanying code for the single-cell multiome analysis post Cell Ranger can be found at this GitHub repository (https://github.com/razofz/DB_QZ_multiome, copy archived at *Olofzon, 2023*), including conda specifications for the environments used with exact versions of all packages.

## Preprocessing

The count matrices were generated with the Cell Ranger ARC tool (v2.0.0), aligned to the reference genome mm10. The aggregated dataset (containing data for both samples) were then loaded into Signac/Seurat in R where non-standard chromosomes were removed and the Seurat object split into the two samples, here named Diverse and Immature, which corresponds to whole culture or sorted EPCR+ cells, respectively. The data were then filtered on UMI, gene and peak counts, as well as nucleosome signal and transcription start site (TSS) enrichment, with separate thresholds for the two samples. The ATAC features were classified as either distal or proximal peaks, and split accordingly. Proximal peaks were defined as a±2 kbp region around the TSS. The TSS positions were retrieved from GREAT's website, and GREAT v4 mm10 version was used. All peaks not classified as proximal received the classification 'distal'. Motifs were then identified for distal and proximal peaks separately (*Fornes et al., 2020*).

## Downstream processing and joint-modality dimensionality reduction

The samples were processed separately. Since SAILERX uses pre-existing dimensionality reduction and clusters for the RNA modality, the RNA data was normalized and scaled, highly variable genes (HVGs) were found, principal component analysis (PCA) was run, and clustering was performed, all with the Seurat standard functions with default parameters. After that, relevant data was extracted (metadata including RNA clusters, PCA embeddings, a whitelist of non-filtered out cell barcodes, ATAC peak count matrix) and with it an hdf5 file similar to the example files provided by the SAILERX repository was created (https://github.com/uci-cbcl/SAILERX; *Cao, 2022*). A SAILERX model was then trained on the generated hdf5 files (still sample-wise) according to their provided instructions. The resulting embeddings from the SAILERX joint-modality dimensionality were then exported and imported into the respective Seurat objects. In Seurat, the embeddings were used for graph construction, clustering, and constructing UMAPs. Scores for cell cycle gene signatures were calculated and visualized (*Figure 2E* for the Immature sample) using Seurat's corresponding standard functions with default parameters.

## Differential gene expression

Differential gene expression testing was performed on the resulting clusters using Seurat's FindAllMarkers function with the HVGs found above. This analysis was used to calculate HSC signature score as described below (section 'HSC signature').

## ATAC processing and cluster annotation

Motif activity scores were identified with chromVAR (*Schep et al., 2017*) for distal and proximal peaks, respectively, still sample-wise. Differentially active motifs (DAMs) were then identified for each cluster. Annotation of clusters in terms of cell type was then performed manually with the help of DEGs and DAMs (*Figure 2B*), guided by pre-existing data on the expression profiles of cell types using the Enrichr platform (https://maayanlab.cloud/Enrichr/).

## HSC signature

A 13 gene condensed HSC signature list (*Figure 2—figure supplement 1C*) was obtained by mining genes associated to cluster 5 with publicly available gene expression data of HSCs from the ImmGen consortia and Bloodspot (https://servers.binf.ku.dk/bloodspot/). This signature was then used to calculate a score with Seurat's AddModuleScore function. A classification for expression of this gene signature was performed with a slightly modified version of Seurat's CellCycleScoring function, where instead of three classifications (S, G2M, or Undecided/G1) two classifications were used (expressed or not expressed). This classification was then used to create the contours used in *Figure 2D–E*, and the signature score used for coloring the points in the same plots.

## Trajectory inference in the Diverse sample

Trajectory inference was performed for the Diverse sample with the slingshot package (*Street et al., 2018*). As initiating cluster, the cluster 5 was assigned (the earliest cluster), and end clusters chosen were clusters 4, 1, 6, and 7. The slingshot function of slingshot was used with otherwise default parameters. A joint UMAP with the four resulting trajectories was constructed (*Figure 2C*). To circumvent the problem of interfering color scales, only the top 20% expressing cells for each marker were chosen for coloring, and overlapping cells between these sets were excluded.

## Assessment of HSC heterogeneity via DNA barcoding

Lentiviral barcode libraries were purchased from AddGene (No. 115645 and No. 83993) and amplified according to instructions (*Biddy et al., 2018*; *Horlbeck et al., 2016*). AddGene 83993 is a Crispri gRNA library that was repurposed here for the purpose of lentiviral barcoding (e.g. no Cas9 was co-expressed). These libraries were kind gifts from Samantha Morris and Jonathan Weissman.

For pre-culture labeling, five wells containing 1000 CD45.1$^+$ HSCs each were sorted and cultured for 48 hr. Cells were then transduced with the AddGene 115645 library (containing >5737 unique barcodes) overnight with approximately 35% transduction efficiency, which was followed by regular culture procedures. After 19 days, expanded cells were collected from each initial wells separately. Half of the expanded cells were transplanted into individual 'parental' CD45.2$^+$ recipients and the rest were mixed together and transplanted into five 'daughter' CD45.2$^+$ recipients. Each 'parental' or 'daughter' recipient received an estimated expansion equivalent to 500 cHSCs. For post-culture labeling, 3000 cHSCs were sorted from CD45.1$^+$ mice and expanded for 13 days. Thereafter, the cells were transduced with the AddGene 83993 library (containing >10,090 unique barcodes) overnight at an approximate density of 100,000 cells/well. Half of the cultured and transduced cells were transplanted into 5 CD45.2$^+$ WT recipients. Each recipient received an estimated expansion equivalent to 300 cHSCs, of which approximately 15% were transduced according to assessments of transduction efficiencies. Barcode-labeled cells were collected and analyzed from four recipients. For both pre- and post-culture labeling experiments, recipients were lethally irradiated and provided with 500,000 unfractionated CD45.2+WBM cells each as support. At the experimental endpoints (16 weeks after transplantation), barcode labeled BM myeloid cells were sorted into 200 µl RNA lysis buffer (Norgen Biotek Corp) ('pre-culture' barcoding) or pelleted for genomic DNA extraction ('post-culture' barcoding) and proceeded to library preparation and sequencing analysis.

RNA was isolated according to the protocol of Single Cell RNA Purification Kit (Norgen Biotek Corp) and cDNA was synthesis using qScript cDNA SuperMix (Quantabio). Genomic DNA was extracted using PureLink Genomic DNA Mini Kit (Invitrogen). The 8- (pre-culture) and 21-base pair (bp) barcodes (post-culture) were amplified by PCR using Q5 High-Fidelity DNA Polymerase (New England Biolabs). Primers used are listed in *Supplementary file 1k*. Adapters containing index sequences (*Supplementary file 1k*) were added by PCR for 7 cycles. After each PCR amplification step, the products were cleaned up using SPRIselect beads (Beckman Coulter). Purified libraries were run on Agilent High Sensitivity DNA Kit chip (Agilent Technologies) to verify the expected size distribution, quantified by Qubit dsDNA HS Assay Kit (Thermo Fisher Scientific), and pooled at equimolar concentrations. Pooled libraries were loaded on an Illumina MiSeq Reagent Nano Kit v2 flow cell following protocols for single-end 100-cycle sequencing. FASTQ files were further assessed with FastQC (v0.11.2), including read count, base quality across reads, and guanine and cytosine content per sequence.

After extracting the reads of each barcode, the frequency in each recipient was calculated and barcodes with a frequency less than 0.1% were considered background reads. This is an arbitrary cutoff. Barcodes present in more than one 'parental' recipient were excluded from analysis (7% and 2% of barcodes pre- and post-culture, respectively) to avoid the possibility of including HSCs transduced independently with the same barcode or potential over-represented barcodes. Reads were normalized to 10$^6$ total reads per recipient followed by statistical analyses.

## CTV labeling for in vitro and in vivo HSC proliferation tracing

CD45.2$^+$-cultured cells, cKit-enriched BM cells, or CD4-enriched spleen cells were collected, washed, and pelleted. Cells were resuspended to a final concentration of 5×10$^6$ cells/ml and labeled with 1 µM CTV (Invitrogen) in PBS for 10 min at 37°C. The reaction was stopped by adding same volume of ice-cold FACS buffer and washing again with FACS buffer. For in vitro proliferation tracing, 10$^5$ CTV-labeled

cKit-enriched BM cells were plated into each well containing F12-PVA culture media (*Supplementary file 1a*). At each time point, half of the cultured cells were collected and stained for evaluating cHSC CTV signals (*Supplementary file 1h*) while the remaining cells were kept in culture for continuous expansion. For in vivo proliferation tracing, CTV-labeled EE100-cultured cells or $5\times10^6$ cKit-enriched BM cells were mixed with $2\times10^6$ CTV-labeled CD4-enriched spleen cells and transplanted into one CD45.1$^+$/CD45.2$^+$ unconditioned recipients. For analysis, BM cells were cKit-enriched and stained accordingly (*Supplementary file 1i*). CD4 positive spleen cells were collected from the same recipient as a control for positive signal/undivided cells (*Supplementary file 1j*).

## Statistical test

Statistical analyses were performed using Microsoft Excel and GraphPad Prism 9. Significance was calculated by Mann-Whitney tests if not otherwise specified. Statistical significances are described in figure legends for the relevant graphs. In all legends, n denotes biological replicates.

## Acknowledgements

We acknowledge Dr. Shabnam Kharazi for scientific discussions and technical support, Mr. Yun Sheng for advice on analysis for barcode sequencing, and Prof. Hiromitsu Nakauchi and Dr. Adam C Wilkinson for advice on cell cultures. The work was supported by grants from the Tobias Foundation, the Swedish Cancer Foundation, Barncancerfonden, and the Swedish Research Council to DB and the Royal Physiographic Society of Lund foundation to QZ and AK-C.

## Additional information

### Funding

| Funder | Grant reference number | Author |
|---|---|---|
| Kungliga Fysiografiska Sällskapet i Lund | | Qinyu Zhang Anna Konturek-Ciesla |
| Tobias Foundation | | David Bryder |
| Cancerfonden | | David Bryder |
| Barncancerfonden | | David Bryder |

The funders had no role in study design, data collection and interpretation, or the decision to submit the work for publication.

### Author contributions

Qinyu Zhang, Conceptualization, Resources, Data curation, Formal analysis, Funding acquisition, Validation, Investigation, Visualization, Methodology, Writing - original draft, Project administration, Writing - review and editing; Rasmus Olofzon, Data curation, Formal analysis, Validation, Visualization, Methodology, Writing - original draft, Writing - review and editing; Anna Konturek-Ciesla, Data curation, Funding acquisition, Validation, Visualization, Methodology; Ouyang Yuan, Data curation, Validation, Visualization, Methodology; David Bryder, Conceptualization, Resources, Supervision, Funding acquisition, Validation, Investigation, Methodology, Writing - original draft, Project administration, Writing - review and editing

### Author ORCIDs

Qinyu Zhang ![ORCID] http://orcid.org/0000-0001-8292-938X
Rasmus Olofzon ![ORCID] http://orcid.org/0000-0001-8079-8718
Anna Konturek-Ciesla ![ORCID] http://orcid.org/0000-0003-1746-930X
David Bryder ![ORCID] http://orcid.org/0000-0002-8761-4237

### Ethics

Work involving animal experimentation had been conducted according to local ethical standards. All experimental procedures were approved by a local ethical committee (permits M186-15 and 16468/2020).

Reviewer #1 (Public Review): https://doi.org/10.7554/eLife.91826.3.sa1
Reviewer #2 (Public Review): https://doi.org/10.7554/eLife.91826.3.sa2
Author Response https://doi.org/10.7554/eLife.91826.3.sa3

## Additional files

### Supplementary files

• Supplementary file 1. Tables for (a) contents of murine hematopoietic stem cell (HSC) media for in vitro culture, (b–j) antibodies used in the study, and (k) primers used in the study.
• MDAR checklist

### Data availability

Sequencing data have been deposited in GEO under accession codes GSE234906. All data generated or analysed during this study are included in the manuscript and supporting files; source data files have been provided for all figures.

The following dataset was generated:

| Author(s) | Year | Dataset title | Dataset URL | Database and Identifier |
|---|---|---|---|---|
| Zhang Q, Olofzon R, Konturek-Ciesla A, Yuan O, Bryder D | 2023 | Ex Vivo Expansion Potential of Murine Hematopoietic Stem Cells: A Rare Property Only Partially Predicted by Phenotype | https://www.ncbi.nlm.nih.gov/geo/query/acc.cgi?acc=GSE234906 | NCBI Gene Expression Omnibus, GSE234906 |

The following previously published datasets were used:

| Author(s) | Year | Dataset title | Dataset URL | Database and Identifier |
|---|---|---|---|---|
| Ali NJ, Montserrat-Vazquez S, Mallm JP, Florian MC | 2022 | Transplanting rejuvenated blood stem cells extends lifespan of aged immunocompromised mice [scRNA-seq of LSK cells] | https://www.ncbi.nlm.nih.gov/geo/query/acc.cgi?acc=GSE197070 | NCBI Gene Expression Omnibus, GSE197070 |
| Wilson N, Diamanti E | 2015 | Molecular signatures of heterogeneous stem cell populations are resolved by linking single cell functional assays to single cell gene expression | https://www.ncbi.nlm.nih.gov/geo/query/acc.cgi?acc=GSE61533 | NCBI Gene Expression Omnibus, GSE61533 |
| Che J, Bode D, Kucinski I, Cull A, Bain F, Barile M, Boyd G, Belmonte M, Rubio-Lara J, Shepherd M, Clay A, Wilkinson AC, Yamazaki S, Göttgens B, Kent DG | 2022 | Tracking ex vivo hematopoietic stem cell function using Fgd5 and EPCR reveals molecular regulators of expansion | https://www.ncbi.nlm.nih.gov/geo/query/acc.cgi?acc=GSE175400 | NCBI Gene Expression Omnibus, GSE175400 |

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
