## [Editor Report · eLife assessment]

This study presents a **valuable** dissection on how functional HSCs are expanded in PVA cultures. The functional and multi-omic analyses provided are **convincing**, although the additional data and their analysis provided during revision could have been included in the test to assist readers and to strengthen the published manuscript. Nevertheless, the present work will be of value for stem cell biologists interested in HSC regulation.

---

## [Referee Report · Reviewer #1 (Public Review)]

In 2019, Wilkinson and colleagues (PMID: 31142833) managed to break the veil on a 20-year open question on how to properly culture and expand Hematopoietic Stem Cells (HSCs). Although this study is revolutionizing the HSC biology field, several questions regarding the mechanisms of expansion remain open. Leveraging on this gap, Zhang et al.; embarked on a much-needed investigation regarding HSC self-renewal in this particular culturing setting.

The authors firstly tacked the known caveat that some HSC membrane markers are altered during in vitro cultures by functionally establishing EPCR (CD201) as a reliable and stable HSC marker (Figure 1), demonstrating that this compartment is also responsible for long-term hematopoietic reconstitution (Figure 3). Next in Figure 2, the authors performed single-cell omics to shed light into the potential mechanisms involved into HSC maintenance, and interestingly it was shown that several hematopoietic populations like monocytes and neutrophils are also present in this culture conditions, which has not been reported. The study goes on to functionally characterize these cultured HSCs (cHSC). The authors elegantly demonstrate using state-of-the-art barcoding strategies that these culturing conditions provoke heterogeneity in the expanding HSC pool (Figure 4). In the last experiment (Figure 5), it was demonstrated that cHSC not only retain their high EPCR expression levels but upon transplantation these cells remain more quiescent than freshly-isolated controls.

Taken together, this study independently validates that the proposed culturing system works and provide new insights into the mechanisms whereby HSC expansion takes place.

Following a first round of comments, the authors provided a comprehensive point-by-point response to the different points raised by reviewers, which significantly helps on better understanding some of the decisions taken by the authors. However, it is surprising that the current manuscript is practically unchanged compared to the previous version. Effectively, all major comments raised by reviewers are address in the response letter rather than incorporated into a truly updated version, which would be of great benefit for readers.

Further comments:

1. It is highly appreciated that the authors provide a comprehensive and cohesive explanations on (i) the rationale for employing SAILERX for single-cell RNA and ATAC-seq, (ii) data on HSC signature projected on independent scRNA-seq datasets and (iii) further context on the Fgd5 expression limitations. These are important snippets of information which do not only further validate this manuscript's data but also provide context within the HSC biology field.

However, I do not fully agree with the author statement "our primary objective in this study was to highlight the relatively low content of HSCs in cultures" (page 1, response to Reviewers) justifying why single-cell genome-wise approaches were used. As the authors are aware HSCs are defined by functional characterization rather than transcriptional/chromatin accessibility profiles, so it seems odd that this was the rationale to perform omics for this purpose. More importantly, the authors had gone through the lengths of already performing this costly and time-consuming experiment, but miss out on the opportunity to take a deeper dive into the molecular characteristics that could explain divergent behavior between freshly-isolated and cultured HSCs. It would be extremely relevant to the HSC biology community to understand, for example, if these two HSC populations have differences in enhancer accessibility (if the data quality allows), which could provide an upstream explanation for differences in transcription (is also not explored in this manuscript version).

2. It intriguing that the authors acknowledge that there are already more recent versions of this expansion protocol (page 2, response to Reviewers) and provided a convoluted explanation on why these were not included in the original manuscript. Both papers (PMID: 36809781 and PMID: 37385251) have now been published in respected peer-reviewed journals and provide insights which are pertinent for this work. Yet, the authors decided not to discuss these findings. It is understandable that repeating experiments with these updated conditions is outside of the scope of this manuscript, but it would be relevant to discuss how these recent advances in the protocol impact the work presented in this manuscript.

3. Regarding the previous comment on how cultured HSC are related to HSC aging, I highly appreciate both data on serial transplantation and also on scRNA-seq.

---

## [Referee Report · Reviewer #2 (Public Review)]

Summary:

In this study, Zhang and colleagues characterise the behaviour of mouse hematopoietic stem cells when cultured in PVA conditions, a recently published method for HSC expansion (Wilkinson et al., Nature, 2019), using multiome analysis (scRNA-seq and scATACseq in the same single cell) and extensive transplantation experiments. The latter are performed in several settings including barcoding and avoiding recipient conditioning. Collectively the authors identify several interesting properties of these cultures namely: (1) only very few cells within these cultures have long-term repopulation capacity, many others however have progenitor properties which can rescue mice from lethal myeloablation; (2) single cell characterisation by combined scRNAseq and scATACseq is not sufficient to identify cells with repopulation capacity; (3) expanded HSCs can be engrafted in unconditioned host and return to quiescence.

The authors also confirm previous studies that EPCRhigh HSCs have better reconstitution capability than EPCRlow HSCs when transplanted.

Strengths:

The major strength of this manuscript is that it describes how functional HSCs are expanded in PVA cultures to a deeper extent that what has been done in the original publication. The authors are also mindful of considering the complexities of interpreting transplantation data. As these PVA cultures become more widely used by the HSC community, this manuscript is valuable as it provides a better understanding of the model and its limitations.

Novelty aspects include:

• The authors determined that small numbers of expanded HSCs enable transplantation into non-conditioned syngeneic recipients.

• This is to my knowledge the first report characterising output of PVA cultures by multiome. This could be a very useful resource to the field.

• They are also the first to my knowledge to use barcoding to quantify HSC repopulation capacity at the clonal level after PVA culture.

• It is also useful to report that HSCs isolated from fetal livers do expand less than their adult counterparts in these PVA cultures.

Weaknesses:

• The analysis of the multiome experiment is limited. The authors do not discuss what cell types, other than functional or phenotypic HSCs are present in these cultures (are they mostly progenitors or bona fide mature cells?) and no quantifications are provided. It seems nonetheless that most cells in these cultures do not acquire differentiation markers. In addition, the functional experiments demosntrate very few retain transplantation capacity. Future works will have to investigate the nature of the bulk of the other cells in these cultures.

• Barcoding experiments are technically elegant but do not bring particularly novel insights.

• Number of mice analysed in certain experiments is fairly low (Figure 1 and 5).

• The manuscript remains largely descriptive. While the data can be used to make useful recommendations to future users working with PVA cultures and in general with HSCs, those recommendations could be more clearly spelled out in the discussion.

• The authors could have provided discussion of the other publications/preprints which have used these methods to date. This would have been useful for researchers who have not used this technique.

Overall, the authors succeeded in providing a useful set of experiments to better interpret what type of HSCs are expanded in PVA cultures. More in depth mining of their bioinformatic data (by the authors or other groups) is likely to highlight other interesting/relevant aspects of HSC biology in relation to this expansion methodology.

---

## [Author Response]

The following is the authors’ response to the original reviews.

**Reviewer #1 (Public Review):**
In 2019, Wilkinson and colleagues (PMID: 31142833) managed to break the veil in a 20-year open question on how to properly culture and expand Hematopoietic Stem Cells (HSCs). Although this study is revolutionizing the HSC biology field, several questions regarding the mechanisms of expansion remain open. Leveraging on this gap, Zhang et al.; embarked on a much-needed investigation regarding HSC self-renewal in this particular culturing setting.The authors firstly tacked the known caveat that some HSC membrane markers are altered during in vitro cultures by functionally establishing EPCR (CD201) as a reliable and stable HSC marker (Figure 1), demonstrating that this compartment is also responsible for long-term hematopoietic reconstitution (Figure 3). Next in Figure 2, the authors performed single-cell omics to shed light on the potential mechanisms involved in HSC maintenance, and interestingly it was shown that several hematopoietic populations like monocytes and neutrophils are also present in this culture conditions, which has not been reported. The study goes on to functionally characterize these cultured HSCs (cHSC). The authors elegantly demonstrate using state-of-the-art barcoding strategies that these culturing conditions provoke heterogeneity in the expanding HSC pool (Figure 4). In the last experiment (Figure 5), it was demonstrated that cHSC not only retain their high EPCR expression levels but upon transplantation, these cells remain more quiescent than freshly-isolated controls.Taken together, this study independently validates that the proposed culturing system works and provides new insights into the mechanisms whereby HSC expansion takes place.Most of the conclusions of this study are well supported by the present manuscript, some aspects regarding experimental design and especially the data analysis should be clarified and possibly extended.1. The first major point regards the single-cell (sc) omics performed on whole cultured cells (Figure 2):a. The authors claim that both RNA and ATAC were performed and indeed some ATAC-seq data is shown in Figure 2B, but this collected data seems to be highly underused.

We appreciate the opportunity to clarify our analytical approach and the rationale behind it.In our study, we employed a novel deep learning framework, SAILERX, for our analysis. This framework is specifically designed to integrate multimodal data, such as RNAseq and ATACseq. The advantage of SAILERX lies in its ability to correct for technical noise inherent in sequencing processes and to align information from different modalities. Unlike methods that force a hard alignment of modalities into a shared latent space, SAILERX allows for a more refined integration. It achieves this by encouraging the local structures of the two modalities, as measured by pairwise similarities.

To put it more simply, SAILERX combines RNAseq and ATACseq data, ensuring that the unique characteristics of each data type are respected and used to enhance the overall biological picture, rather than forcing them into a uniform framework.

While it is indeed possible to analyze the ATAC-seq and RNA-seq modalities separately, and we acknowledge the potential value in such an approach, our primary objective in this study was to highlight the relatively low content of HSCs in cultures. This finding is a key point of our work, and the multiome data support this from a molecular point of view.

The Seurat object we provide was created to facilitate further analysis by interested researchers. This object simplifies the exploration of both the ATAC-seq and RNA-seq data, allowing for additional investigations that may be of interest to the scientific community.We hope this explanation clarifies our methodology and its implications.

b. It's not entirely clear to this reviewer the nature of the so-called "HSC signatures"(SF2C) and why exactly these genes were selected. There are genes such as Mpl and Angpt1 which are used for Mk-biased HSCs. Maybe relying on other HSC molecular signatures (PMID: 12228721, for example) would not only bring this study more into the current field context but would also have a more favorable analysis outcome. Moreover reclustering based on a different signature can also clarify the emergence of relevant HSC clusters.

In our study, the selection of the HSC signature in our work was based on well-referenced datasets on well-defined HSPCs, as detailed in the "v. HSC signature" section of our methods. This signature was projected also to another single-cell RNA sequencing dataset generated from ex vivo expanded HSC culture (PMID: 35971894, see Author response image 1 below), demonstrating again an association primarily to the most primitive cells (at least based on gene expression).

**Author response image 1. sa3fig1:** Projection of "our" HSC signature on scRNAseq data from independent work.

In further response to the suggestion here, we have also examined the molecular signature of HSCs referenced in PMID: 12228721 but also of another HSC signature from PMID: 26004780 in our data (Author response image 2). While these signatures do indeed enrich for cells that fall in the cluster of molecularly defined HSCs, our analysis indicates that neither of them significantly improves the identification of HSCs in our dataset compared to the signature we originally used. This finding reinforces our confidence in the appropriateness of our chosen HSC signature for this study.

**Author response image 2. sa3fig2:** Projection of alternative HSC signatures onto the SAILERX UMAP.

Regarding the specific genes Mpl and Angpt1, we respectfully oppose the view that these genes are exclusively associated with MK-biased HSCs. There is substantial evidence supporting the broader role of Mpl in regulating HSCs, regardless of any particular "lineage bias". Similarly, while Angpt1 has been less extensively studied, its role in HSCs, as examined in PMID: 25821987, suggests a more general association with HSCs rather than a specific impact on MKs. Therefore, we maintain that it is more accurate to consider these genes as HSC-associated rather than restricted to MK-biased HSCs.

Finally, addressing the comment on reclustering based on different signatures, we would like to clarify that the clustering process is independent of the projection of signatures. The clustering aims to identify cell populations based on their overall molecular profiles, and while signatures can aid in characterizing these populations, they do not influence the clustering process itself.

c. The authors took the hard road to perform experiments with the elegant HSC-specific Fgd5-reporter, and they claim in lines 170-171 that it "failed to clearly demarcate in our single-cell multimodal data". This seems like a rather vague statement and leads to the idea that the scRNA-seq experiment is not reliable. It would be interesting to show a UMAP with this gene expression regardless and also potentially some other HSC markers.

We understand the concerns raised about our statement on the performance of the Fgd5-reporter in our multimodal data analysis. Our aim was not to suggest that single-cell molecular data are unreliable. Instead, we intended to point out specific challenges associated with scRNA sequencing, notably the high rates of dropout. Regarding the specific example of Fgd5, it appears this transcript is not efficiently captured by 10x technology. Our previous 10x scRNA-seq experiments on cells from the Fgd5 reporter strain (Säwén et al., eLife 2018; Konturek-Ciesla et al., Cell Rep. 2023) support this observation. Despite cells being sorted as Fgd5-reporter positive, many showed no detectable transcripts.

We consider it pertinent to note that our study integrates ATAC-seq data in conjunction with single-cell molecular data. We believe that this integration, coupled with the analytical methods we have employed, potentially offers a way to address some of the limitations typically associated with scRNA sequencing. However, in assessing frequencies, we observe that the number of candidate HSCs identified via single-cell molecular data is substantially higher compared to those identified through flow cytometry, the latter which we demonstrate correlate functionally with genuine long-term repopulating activity.

With respect to Fgd5, as depicted in our analysis below, there appears to be an enrichment of cells in the cluster identified as HSCs, as well as a significant representation in the cycling cell cluster (Author response image 3). Regarding the projection of other individual genes, the Seurat object we have provided allows for such projections to be readily performed. This offers an opportunity for further exploration and validation of our findings by interested researchers.

**Author response image 3. sa3fig3:** Feature plot depicting Fgd5 expression in the SAILERX UMAP.

1. During the discussion and in Figure 4, the authors ponder and demonstrate that this culturing system can provoke divert HSC close expansion, having also functional consequences. This a known caveat from the original system, but in more recent publications from the original group (PMID: 36809781 and PMID: 37385251) small alterations into the protocol seem to alleviate clone selection. It's intriguing why the authors have not included these parameters at least in some experiments to show reproducibility or why these studies are not mentioned during the discussion section.

Thank you for pointing out the recent publications (PMID: 36809781 and PMID: 37385251) that discuss modifications to the HSC culturing system. We appreciate the opportunity to address why these were not included in our discussion or experiments.

Firstly, it is important to note that these papers were published after the submission of our manuscript. In fact, one of the studies (PMID: 36809781) references the preprint version of our work on Biorxiv. This timing meant that we were unable to consider these studies in our initial manuscript or incorporate any of their findings into our experimental designs.

Furthermore, as strong advocates for the peer-review system, we prioritize references that have undergone this rigorous process. Preprints, while valuable for early dissemination of research findings, do not offer the same level of scrutiny and validation as peer-reviewed publications. Our approach was to rely on the most relevant and rigorously reviewed literature available to us at the time of submission. This included, most notably, the original and ground-breaking work by Wilkinson et al., which provided a foundational basis for our research.

We acknowledge that the field of HSC research is rapidly evolving, and new findings, such as those mentioned, are continually emerging. These new studies undoubtedly contribute valuable insights into HSC culturing systems and their optimization. However, given the timing of their publication relative to our study, we were not able to include them in our analysis or discussion.

1. In this reviewer's opinion, the finding that transplanted cHSC are more quiescent than freshly isolated controls is the most remarkable aspect of this manuscript. There is a point of concern and an intriguing thought that sprouts from this experiment. It is empirical that for this experiment the same HSC dose is transplanted between both groups. This however is technically difficult since the membrane markers from both groups are different. Although after 8 weeks chimerism levels seem to be the same (SF5D) for both groups, it would strengthen the evidence if the author could demonstrate that the same number of HSCs were transplanted in both groups, likely by limiting dose experiments. Finally, it's interesting that even though EE100 cells underwent multiple replication rounds (adding to their replicative aging), these cells remained more quiescent once they were in an in vivo setting. Since the last author of this manuscript has also expertise in HSC aging, it would be interesting to explore whether these cells have "aged" during the expansion process by assessing whether they display an aged phenotype (myeloid-skewed output in serial transplantations and/or assisting their transcriptional age).

We thank the reviewer for the insightful observations regarding the quiescence of transplanted cultured HSCs. We appreciate the opportunity to clarify the experimental design and its implications, particularly in the context of HSC aging.

The primary aim of comparing cKit-enriched bone BM cells with cultured cells was to investigate if ex vivo activated HSCs exhibit a similar proliferation pattern to in vivo quiescent HSCs post-transplantation. This comparison was crucial for evaluating the similarity between in vitro cultured and "unmanipulated" HSC behavior. While we acknowledge the technical challenge of transplanting equivalent HSC doses between groups due to differing membrane markers, our study design focused on assessing stem cell activity post-culture. This was quantitatively evaluated by calculating the repopulating units (detailed in Table 1 and Fig S4G), rather than through a limiting dilution assay. There exists a plethora of literature demonstrating the correlation between these assays, although of course the limiting dilution assay is designed to provide a more exact output.

Regarding the intriguing aspect of HSC aging in the context of ex vivo expansion, our observations indicate that both the subfraction of ex vivo expanded cells (Fig 3 and Fig S3) and the entire cultured population (Fig 4B, Fig 5B, Fig S4A, and Fig S5B) maintain long-term multilineage reconstitution capacity post-transplantation. This suggests that the PVA-culture system does not lead to apparent signs of "HSC aging," despite the cells undergoing active self-renewal in vitro. This is further supported by our serial transplantation experiments, where cultured cells continued to demonstrate multilineage capacity rather than any evident myeloid-biased reconstitution 16 weeks post-second transplantation (see Author response image 4 below).

**Author response image 4. sa3fig4:** Serial transplantation behavior of ex vivo expanded HSCs. 5 million whole BM cells from primary transplantation were transplanted together with 5 million competitor whole BM cells. The control group was transplanted with 100 cHSCs freshly isolated from BM for the primary transplantation. Mann-Whitney test was applied and the asterisks indicate significant differences. *, p < 0.05; **, p < 0.01; ****, p < 0.0001. Error bars denote SEM.

However, we recognize the complexity of defining HSC aging and the potential for the culture system to influence certain aspects of this process. The association of aging signature genes with HSC primitiveness and young signature genes with differentiation presents an interesting dichotomy. Our analysis of a native dataset on young mice and the projection of aged signatures onto our multiome data (as shown below for a set of genes known to be induced at higher levels in aged HSCs (f.i. Wahlestedt et al., Nature Comm 2017), aging scRNAseq data from PMID: 36581635) does not directly indicate that the culture system promotes HSC aging compared to aged Lin-Sca+Kit+ cells. Yet, we do not rule out the possibility that culturing may influence other facets of the HSC aging process.

In conclusion, while our current data do not provide direct evidence of induced HSC aging through the culture system, this remains a compelling area for future research. The potential impact of ex vivo culture on aspects of the HSC aging process warrants further exploration, and we appreciate your suggestion in this regard.

**Author response image 5. sa3fig5:** No evident signs of "molecular aging" following ex vivo expansion of HSCs. Young and aged scRNAseq data from PMID: 36581635 were integrated and explored from the perspective of known genes associating to HSC aging. The top row depicts contribution to UMAPs from young and aged cells (two left plots), cell cycle scores of the cells, and the expression of EPCR and CD48 as examples markers for primitive and more differentiated cells, respectively. The expression of the HSC aging-associated genes Wwtr1, Cavin2, Ghr, Clu and Aldh1a1 was then assessed in the data as well as in the SAILERX UMAP of cultured HSCs (bottom row).

**Reviewer #2 (Public Review):**
Summary:In this study, Zhang and colleagues characterise the behaviour of mouse hematopoietic stem cells when cultured in PVA conditions, a recently published method for HSC expansion (Wilkinson et al., Nature, 2019), using multiome analysis (scRNA-seq and scATACseq in the same single cell) and extensive transplantation experiments. The latter are performed in several settings including barcoding and avoiding recipient conditioning. Collectively the authors identify several interesting properties of these cultures namely: (1) only very few cells within these cultures have long-term repopulation capacity, many others, however, have progenitor properties that can rescue mice from lethal myeloablation; (2) single-cell characterisation by combined scRNAseq and scATACseq is not sufficient to identify cells with repopulation capacity; (3) expanded HSCs can be engrafted in unconditioned host and return to quiescence.The authors also confirm previous studies that EPCRhigh HSCs have better reconstitution capability than EPCRlow HSCs when transplanted.Strengths:The major strength of this manuscript is that it describes how functional HSCs are expanded in PVA cultures to a deeper extent than what has been done in the original publication. The authors are also mindful of considering the complexities of interpreting transplantation data. As these PVA cultures become more widely used by the HSC community, this manuscript is valuable as it provides a better understanding of the model and its limitations.Novelty aspects include:• The authors determined that small numbers of expanded HSCs enable transplantation into non-conditioned syngeneic recipients.• This is to my knowledge the first report characterising the output of PVA cultures by multiome. This could be a very useful resource for the field.• They are also the first to my knowledge to use barcoding to quantify HSC repopulation capacity at the clonal level after PVA culture.• It is also useful to report that HSCs isolated from fetal livers do expand less than their adult counterparts in these PVA cultures.Weaknesses:• The analysis of the multiome experiment is limited. The authors do not discuss what cell types, other than functional or phenotypic HSCs are present in these cultures (are they mostly progenitors or bona fide mature cells?) and no quantifications are provided.

The primary objective of our manuscript was to characterize the features of HSCs expanded from ex vivo culture. In this context, our analysis of the single cell multiome sequencing data was predominantly centered on elucidating the heterogeneity of cultures, along with subsequent in vivo functional analysis. This focus is reflected in our comparisons between the molecular features of ex vivo cultured candidate HSCs (cHSCs) and "fresh/unmanipulated" HSCs, as illustrated in Figures 2D-E of our manuscript.

Our findings provide substantial evidence that ex vivo expanded cells share significant similarities with HSCs isolated from the BM in terms of molecular features, differentiation potential, heterogeneity, and in vivo stem cell activity/function. This suggests that the ex vivo culture system closely mimics several aspects of the in vivo environment, thereby broadening the potential applications of this system for HSC research.

Regarding the presence of other cell types in the cultures, it is important to note that most cells did not express mature lineage markers, suggesting their immature status. However, we acknowledge the presence of some mature lineage marker-positive cells within the cultures. These cells are represented by the endpoints in our SAILERX UMAP, indicating a progression from immature to more differentiated states within the culture system.

While the main emphasis of our study was on HSCs, we understand the importance of acknowledging and briefly discussing the presence and characteristics of other cell types in the cultures. This aspect provides a more comprehensive understanding of the culture system and its impact on cellular heterogeneity, although it was for the most part beyond the scope of our studies.

• Barcoding experiments are technically elegant but do not bring particularly novel insights.We respectfully disagree with the view that our barcoding experiments do not offer novel insights. We believe that the application of barcoding technology in our study represents a significant advancement over previous methods, both in terms of quantitative rigor and ethical considerations.

In the foundational work by Wilkinson et al., clonal assessments were indeed performed, but these were limited in scope and largely served as proof of concept. Our use of barcoding technology, on the other hand, allowed for a comprehensive quantitative assessment of the expansion potential of HSC clones. This technology enabled us to rigorously quantify the number of HSC clones capable of undergoing at least three self-renewing divisions (e.g. those clones present in 5 separate animals), while also revealing the heterogeneity in their expansion potential.

One alternative approach could have been to culture single HSCs and distribute the progeny among multiple mice for analysis. However, when considering the sheer number of mice that would be required for such an experiment for quantitative assessments, it becomes evident that viral barcoding is a far superior method. Not only does it provide a more efficient and scalable approach to assessing clonal expansion, but it also significantly reduces the number of animals required for the study, aligning with the principles of ethical research and animal welfare.

In conclusion, we assert that the barcoding experiments conducted in our study are not only technically robust but also yield novel quantitative insights into the dynamics of HSC clones within expansion cultures. These insights have value not only for current research but also hold potential implications for future applications.

• The number of mice analysed in certain experiments is fairly low (Figures 1 and 5).

We would like to clarify our approach in the context of the 3R (replacement, refinement, and reduction) policy, which guides ethical considerations in animal research.

In alignment with the 3R principles, our study was designed to minimize the use of experimental animals wherever possible. For most experiments, including those presented in Figures 1 and 5, we adopted a standard of using five mice per group. Based on the effect sizes we observed, we concluded that this sample size was appropriate for most parts of our study.

Specifically for Figure 5, we used two animals per time point, totaling seven animals per treatment group. It is important to note that we did not monitor the same animals over time but used different animals at each time point, as mice had to be sacrificed for the type of analyses conducted. Despite the seemingly small sample size, the results we obtained were remarkably consistent across groups. This consistency provided strong evidence that ex vivo activated HSCs return to a more quiescent state after being transplanted into unconditioned recipients. Given the clear and consistent nature of these results, we determined that including more animals for the purpose of additional statistical analysis was not necessary.

Our approach reflects a balance between adhering to ethical standards in animal research and ensuring the scientific validity and reliability of our findings. We believe that the sample sizes chosen for our experiments are justified by the consistent and significant results we obtained, which contribute meaningfully to our understanding of HSC behavior post-transplantation.

• The manuscript remains largely descriptive. While the data can be used to make useful recommendations to future users working with PVA cultures and in general with HSCs, those recommendations could be more clearly spelled out in the discussion.

We fully agree that many aspects of our study are indeed descriptive, which is reflective of the exploratory and foundational nature of this type of research.

We have strived to provide clear and direct recommendations for researchers interested in utilizing the PVA culture system, which we believe are evident throughout our manuscript:

1. Utility of Viral Delivery in HSC Research: Our research, particularly through the use of barcoding experiments, underscores the effectiveness of viral delivery methods in HSC studies. While barcoding itself is a significant tool, it is the underlying process of viral delivery that truly exemplifies the potential of this approach. Our work shows that the culture system is highly conducive to maintaining HSC activity, which is critical for genetic manipulation. This is evident not only in our current study but also in our previous work that included for transient delivery methods (Eldeeb et al., Cell Reports 2023).

2. Non-conditioned transplantation: Our findings suggest that non-conditioned transplantation can be a valuable method in studying both normal and malignant hematopoiesis. This approach can complement genetic lineage tracing models, providing a more native and physiological context for hematopoietic research. We state this explicitly in our discussion.

3. Integration with recent technical advances: The combination of the PVA culture system with recent developments in transplantation biology, genome engineering, and single-cell technologies holds significant promise. This integration is likely to yield exciting discoveries with relevance to both basic and clinically oriented hematopoietic research. This is the end statement of our discussion.

While our manuscript is in a way tailored to those with experience in HSC research, we have made a concerted effort to ensure that the content is accessible and informative to a broader audience, including those less familiar with this area of study. Our intention is to provide a resource that is both informative for experts in the field and approachable for newcomers.

• The authors should also provide a discussion of the other publications that have used these methods to date.

We would like to clarify that the scope of literature on the specific methods we employed, particularly in the context of our research objectives, is not extensive. Most of the existing references on these methods come from a relatively narrow range of research groups. In preparing our manuscript, we tried to be comprehensive yet selective in our citations to maintain focus and relevance. Our referencing strategy was guided by the aim to include literature that was most directly pertinent to our study's methodologies and findings.

Overall, the authors succeeded in providing a useful set of experiments to better interpret what type of HSCs are expanded in PVA cultures. More in-depth mining of their bioinformatic data (by the authors or other groups) is likely to highlight other interesting/relevant aspects of HSC biology in relation to this expansion methodology.

We are grateful for the overall positive assessment of our work and the recognition of its contributions to understanding HSC expansion in PVA cultures.

We agree that every study, including ours, has its limitations, particularly regarding the scope and depth of exploration. It is challenging to cover every aspect comprehensively in a single study. Our research aimed to provide a foundational understanding of HSCs in PVA cultures, and we are pleased that this goal appears to have been met.

We also concur with your point on the potential for further in-depth mining of our bioinformatic data. Our hope is that this data can serve as a resource (or at least a starting point) for other investigators.

In conclusion, we hope that our responses have adequately addressed your queries and clarified any concerns. We are committed to contributing to the growth of knowledge in HSC research and look forward to the advancements that our study might enable, both within our team and the wider scientific community.

**Reviewer #1 (Recommendations For The Authors):**
1. In Line 150, the R packages can/should be mentioned just in the method section;

We have moved this text to the methods section.

1. In Figure F3C adding a legend next to the plot would assist the reader in identifying which populations are referred to, as the same color pellet is used for other panels;

We have now adjusted the figure legend position to make it more clear for the reader.

1. In Figure 4D, for the pre-culture experiments 1000 cHSCs were used and then in the post-culture 1200 cHSCs were used. Can the authors justify the different numbers?

The decision to use 1000 cHSCs in the pre-culture experiments and 1200 cHSCs in the post-culture experiments was not based on a specific rationale favoring one cell number over the other. In our Method section, we have detailed our experimental design, which was structured to provide robust and reliable readouts of HSC behavior and characteristics in different conditions.

We consider the two cell numbers – 1000 and 1200 – to be quite similar in the context of our experimental aims. Since the readouts here are based on clonal assessments, this slight difference in cell numbers is unlikely to significantly impact the overall conclusions drawn from these experiments. The primary focus of our study was on qualitative aspects of HSC behavior and function, rather than on quantitative differences that might arise from small variations in initial cell numbers.

1. In SF5F it would help readers if a line plot (per group) was also shown together with the dot plots. Moreover, applying statistics to the trend lines (Wilcoxon, for example) would strengthen the argument that cHSCs divide less than control cells.

We would like to clarify that the data presented in SF5F were derived from different animals at each respective time point. As such, the data points at each time point represent independent measurements from separate animals, rather than a continuous measurement from the same set of animals over time. Therefore, creating a line plot that connects each time point within a group would inadvertently convey a misleading impression of a longitudinal study on the same animals, which is not reflective of the actual experimental design. Instead, the dot plot format was chosen as it more accurately depicts the independent and discrete nature of the measurements at each time point. Our current data presentation method was selected to provide the most accurate and transparent representation of our findings.

**Reviewer #2 (Recommendations For The Authors):**
Listed below are recommendations to further improve this manuscript:Major Comments1. Fig 1: the authors showed that EPCRhigh HSCs have better reconstitution capability than EPCRlow HSCs via bone marrow transplantation. Additionally, mice receiving cultured EPCRhigh SLAM LSK cells were more efficiently radioprotected than those receiving PVA expanded EPCRlow SLAM LSK.a. In addition to Fig.1F, authors should show the lineage distributions and chimerism of mice receiving cultured EPCRhigh and EPCRlow SLAM LSK respectively.

We have indeed analyzed the lineage distribution in these experiments, and our findings indicate no statistically significant differences between the groups (see graph in Author response image 6). This suggests that the cultured EPCRhigh and EPCRlow SLAM LSK cells do not preferentially differentiate into specific lineages in a way that would impact the overall interpretation of our results.

**Author response image 6. sa3fig6:** 

Regarding the chimerism in peripheral blood (PB) lineages, Fig. 1F in our manuscript currently shows the PB myeloid chimerism. We chose to focus on this parameter as it most directly relates to our study's objectives. We did here not transplant with competitor cells, and in most cases, the chimerism levels reached 100% for lineages other than T cells (T cells being more radioresistant). Based on our analysis, including data on chimerism in other PB lineages would not significantly enhance the understanding of the functional capacity of the transplanted cells, as the myeloid chimerism data already provides a robust indicator of their engraftment and functional potential.

We believe that our current presentation of data in Fig. 1F, along with the additional analyses provided in the results section, offers a comprehensive understanding of the behavior and potential of the cultured EPCRhigh and EPCRlow SLAM LSK cells.

b. Fig1F: only 5 mice were used in each group. Could this result occur by chance? Testing with Fisher's exact test with the data provided results in p=0.16. The authors should consider adding more animals or adding the p-value above (or from another relevant test) for readers' consideration.

We acknowledge the point that only five mice were used in each group and understand the concern regarding the robustness of our findings.

As correctly noted, applying Fisher's exact test to the data in Fig. 1F results in a p-value which does not reach the conventional threshold for statistical significance. However, one might also consider the analysis of the KM survival curve, which associated with a p-value of 0.0528 (Fig. 1F, left graph below; Gehan-Breslow-Wilcoxon test). A similar test on the single-cell culture transplantation experiment (Fig. 1E, right graph below) also demonstrated statistical significance (p-value = 0.0485).

While these p-values meet (or are very close to) the conventional criteria for statistical significance (p<0.05), we have chosen to place greater emphasis on effect sizes rather than strictly on p-values. This decision is based on our belief that effect sizes provide a more direct and meaningful measure of the biological impact observed in our experiments. We find that the effect sizes observed are compelling and consistent with the overall narrative of our study.

**Author response image 7. sa3fig7:** 

1. The characterisation of the multiome experiment is highly underdeveloped.a. From an experimental point of view, it is not clear how the PVA culture for this experiment was started. Are there technical/biological replicates? Have several PVA cultures been pooled together?

We have included these details in the revised text to ensure a comprehensive understanding of our experimental setup.

b. Fig2B: The authors should present more data as to how each of the clusters was annotated (bubble plot of marker genes used for annotation?) and importantly the percentage of cells in each of the clusters. It is particularly relevant to note what % is the cluster annotated as HSCs and compare that to the % of phenotypic HSCs and the % repopulating HSCs calculated in the transplantation experiments.

In our study, the annotation of clusters was primarily based on reference genes for cell types from prior works in the field, such as from our recent work (Konturek-Ciesla et al., Cell Reports 2023). Additionally, we employed transcription factor (TF) motifs to assign identities to these clusters. This approach is relatively standard in the field, and we believe it provides a robust framework for our analysis. We included information on some of the key TF motifs used to guide our annotations.

Regarding the assignment of a percentage to cells within the HSC cluster, we initially had reservations about the utility of this measure. This is because the transcriptional identity of HSCs might not align precisely with their identity based on candidate HSC protein markers. There are complexities related to transcriptional continuums that could influence the interpretation of such data. However, acknowledging your request for this information, we have now included the percentage of cells in the HSC cluster in Fig. 2B for reference.

We also wish to highlight that when isolating EPCR+ cells, which encompasses a range of CD48 expression, clustering becomes much less distinct, as shown in Fig. 2E. Most of these cells do not demonstrate long-term functional HSC activity in a transplantation setting (as presented in Figure 3). This observation underscores the challenges in deducing HSC identity based solely on molecular data and reinforces the importance of functional validation.

c. Are there any mature cells in these PVA cultures? The annotations presented in the table under the UMAP are vague: Are cluster 4 monocytes or monocytes progenitors? Same for clusters 0,1 and 7 - are these progenitors or more mature cells? How were HPCs (cluster 3) distinguished from cHSCs (cluster 5)?

We agree with your observation that the annotations for certain clusters, such as clusters 4, 0, 1, and 7, as well as the distinction between HPCs (cluster 3) and cHSCs (cluster 5), appear vague. This vagueness to some extent stems from the challenges inherent in comparing cultured cells to their counterparts isolated directly from animals. Most reference data defining cell types are derived from cells in their native state, and less is known about how these definitions translate to the progeny of HSPCs cultured in vitro.

In our study, we used the expression of reference genes and enriched transcription factor motifs to annotate clusters. This method, while useful, has its limitations in precisely defining the maturation stage of cells in culture. The enrichment of lineage-defining factors at the ends of the UMAP suggests the presence of more mature cells, whereas the lack of lineage marker expression in the majority of cells implies a general lack of terminal differentiation.

This issue is not necessarily unique to the culture situation, as similar challenges in cell type annotation are encountered in other contexts, such as the analysis of granulocyte-macrophage progenitors in bone marrow, where a vast range of cell types and clusters are identified (e.g., PMID: 26627738). To try to address these challenges, we employed an approach detailed in the methods section under the header "iv. ATAC processing and cluster annotation." We assessed marker genes for clusters using Enrichr for cell types, relying on databases designed to provide gene expression identities to defined cell types. This methodology informed our references to the clusters.

In summary, while our annotations provide a general overview of the cell types present in the cultures, we acknowledge the complexities and limitations in precisely defining these types, particularly in distinguishing between progenitors and more mature cells. We hope this explanation clarifies our approach and the considerations behind our cluster annotations, but at the same time feel that the alternative approaches have their own drawbacks.

d. What is the meaning of the trajectories presented in Figure 2C? In the absence of a comparison to (i) what is observed either when HSCs are cultured in control/non-expanding conditions (ii) an in vivo landscape of differentiation in mouse bone marrow; this analysis does not bring any relevant piece of information.

We understand the perspective on comparisons to control conditions and in vivo differentiation landscapes. However, we respectfully disagree with the viewpoint that the analysis that we have performed does not bring relevant information.

The trajectory analysis in Figure 2C is intended to provide insights into the cell types generated in our PVA cultures and the potential differentiation pathways they may follow. This kind of analysis is particularly valuable in the context of understanding how in vitro cultures can support HSC maintenance and differentiation, which is a topic of significant interest in the field. For instance, studies like PMID: 31974159 have highlighted the importance of combining in vitro HSC cultures with molecular investigations.

While we acknowledge that our analysis would benefit from a direct comparison to control or non-expanding conditions, as well as to an in vivo differentiation landscape, we believe that the information provided by our current analysis still holds substantial value. It offers a glimpse into the possible cellular dynamics and differentiation routes within our culture system, which can be a valuable reference point for other investigators working with similar systems.

Regarding the confidence in computed differentiation trajectories, we recognize that this is an area where caution is warranted. Computational approaches to define cell differentiation pathways have inherent limitations and should be interpreted within the context of their assumptions and the data available. This challenge is not unique to our work but is a broader issue in the field of computational biology.

In conclusion, while we agree that additional comparative analyses could further enrich our findings, we maintain that the trajectory analysis presented in Figure 2C contributes meaningful insights into cell differentiation in our PVA culture system. We believe these insights are of interest and value to researchers exploring the complex interplay of HSC maintenance and differentiation in vitro.

1. The addition of barcoding experiments is appreciated. However, it is already known that upon transplantation clonal output is highly heteroegeneous, with a small number of clones predominating over others. This is particularly the case after myeloablation conditioning.a. The "pre-culture" experimental design makes sense. The "post-culture" one is however ambiguous in terms of result interpretation. The authors observe fewer clones contributing to a large proportion of the graft (>5%) than in the "pre-culture" setting. Their interpretation is that expanded HSCs are functionally more homogeneous than the input HSCs. However, in the pre-culture experiment, there are 19 days of expansion during which there will be selection pressures over culture plus ongoing differentiation. In the post-culture experiment, there is no time for such pressures to be exerted. Therefore the conclusion drawn by the authors is not the only conclusion. I would encourage the authors to compare the "pre-culture" experiment to an experiment in which cHSCs are in culture for 48h, then barcoded, and then transplanted. This would be much more informative and would allow a proper comparison of expanded HSCs vs input HSCs.

We understand the perspective that a shorter culture period would reduce the influence of selection pressures and differentiation, potentially allowing for a more direct comparison between expanded HSCs and input HSCs. However, we would like to point out that similar experiments have been conducted in the past, as referenced in our work (PMID: 28224997) and others (PMID: 21964413). These studies have demonstrated a significant heterogeneity in the reconstituting clones when barcoding is done early and cells are transplanted directly.

In light of previous research, we are confident that our methodology — tracking the fates of candidate HSC clones throughout the culture period and assessing the outcomes of individual cells from these expanding clones — yields significant and pertinent insights. We want to highlight the significance of barcoding cells late in the culture, a strategy that allows us to barcode cells that have already been subjected to potential selection pressures within the culture environment. Our primary objective is to investigate the effects of these selection pressures on the subsequent in vivo behavior of the cells that emerge from this process. By focusing on this aspect, we aim to deepen the understanding of how in vitro culture conditions influence the functional characteristics and heterogeneity of HSCs after expansion. We believe this approach provides a unique perspective on the adaptive changes HSCs undergo during culture and their implications for transplantation efficacy and HSC biology. Our study thus addresses a critical question in the field: how do the conditions and selection pressures inherent to in vitro culture impact the quality and behavior of HSCs upon their return to an in vivo environment?

b. Another experiment the authors may consider is barcoding in unconditioned recipients as there the bottleneck of selecting specific clones should be lower. In addition, this could nicely complement the return to quiescence observed in Figure 5 (see point below)

We agree that this experiment could provide valuable insights, particularly in understanding how different selection pressures might affect HSC clones in various transplantation contexts. It would indeed be a worthwhile complement to our observations in Figure 5 regarding the return to quiescence of HSCs post-transplantation.

However, we would like to point out that our study already includes a substantial amount of data and analyses aimed at addressing specific research questions within this defined scope. The addition of an experiment with barcoding in unconditioned recipients, while undoubtedly relevant and interesting, would extend beyond the boundaries we set for this particular study.

1. Figure 5D-F, only 2 animals per condition were tested, so the experiment is underpowered for any statistics. How about cell viability of cHSC after in vitro culture? The authors have also not tested whether there is a difference in cell viability post-transplant between EE100 and control. In addition, comparing cell cycle profiles of donor EPCR+ HSCs in these transplanted mice would provide additional evidence to support the conclusion.

Regarding the sample size, we acknowledge that only two animals per condition were used in these experiments, which limits the statistical power for robust quantitative analysis. This decision was guided by ethical considerations to minimize animal use, in line with the 3Rs principle (Replacement, Reduction, Refinement). Despite the small sample size, we believe that the strong trends observed in these experiments are indicative and consistent with our broader findings, although we recognize the limitations in terms of statistical generalization. At the same time, as we have written in the public response: "Specifically for Figure 5, we used two animals per time point, totaling seven animals per treatment group. It is important to note that we did not monitor the same animals over time but used different animals at each time point, as mice had to be sacrificed for the type of analyses conducted."

In the context of post-transplant analysis, conducting separate viability assessments on transplanted cells is not typically informative. This is because non-viable cells would naturally be eliminated through biological processes such as phagocytosis soon after transplantation. Therefore, any post-transplant viability analysis would not provide meaningful insights into the engraftment potential or behavior of the transplanted cells.

However, it is important to note that in all our cell isolation and analysis protocols, we routinely include viability markers. This practice ensures that the cell populations we study and report on are indeed viable. Including these markers is a standard part of our methodology and contributes to the accuracy and reliability of our data.

Regarding the comparison of cell cycle profiles, we chose to focus on the cell trace assay as a means to monitor and track cell division history, which directly addresses the central theme here - informing on the proliferation and quiescence dynamics of transplanted HSCs. While comparing cell cycle profiles could perhaps offer an additional layer of information, we did not deem it essential for our core objectives.

1. Several publications have used these PVA cultures and made comments on their strengths and limitations. They do not overlap with this study but should be discussed here for completeness (for example Che et al, Cell Reports, 2022; Becker et al., Cell Stem Cell, 2023; Igarashi, Blood Advances, 2023).

See comments to reviewer 1.

Minor CommentsFigure 1C: should add in the legend that this is in peripheral blood.Figure 2C: typo in the title.Figure 3A: typo in "equivalent".We thank the reviewer for catching these errors, which we have now corrected.Figure 3B and 3C: symbol colours of EPCRhighCD48+ and EPCR- are too similar to distinguish the 2 groups easily. We highly recommend using contrasting colours.

For easier visualization, we have changed the symbol types and colors in our revised version.

Fig3B and S3A-B: authors should show statistical significance in comparing the 4 fractions.We have now added this information.In the discussion, the authors rightly point out a paper that described EPCR+ HSCs. There are other papers that also looked at EPCR intensity (high vs low), for example, Umemoto et al., EMBO J, 2022.

While we acknowledge the relevance of the paper you mentioned, we faced constraints in the number of references we could include. Therefore, we prioritized citing the original demonstration of EPCR as an HSC marker, particularly focusing on the work by the Mulligan laboratory, which established that cells expressing the highest levels of EPCR exhibit the most potent HSC activity. We believe this reference most directly supports the core focus of our study and provides the necessary context for our findings.